# The Role of Deep Learning Regularizations on Actors in Offline RL

### Abstract

Deep learning regularization techniques, such as *dropout*, *layer normalization*, or *weight decay*, are widely adopted in the construction of modern artificial neural networks, often resulting in more robust training processes and improved generalization capabilities. However, in the domain of *Reinforcement Learning* (RL), the application of these techniques has been limited, usually applied to value function estimators (Hiraoka et al., 2021; Smith et al., 2022), and may result in detrimental effects. This issue is even more pronounced in offline RL settings, which bear greater similarity to supervised learning but have received less attention. Recent work in continuous offline RL (Park et al., 2024) has demonstrated that while we can build sufficiently powerful critic networks, the generalization of actor networks remains a bottleneck. In this study, we empirically show that applying standard regularization techniques to actor networks in offline RL actor-critic algorithms yields improvements of 6% on average across two algorithms and three different continuous D4RL domains.

## 1  Introduction

Building modern Deep Learning (DL) systems often involves the application of various regularization techniques, such as *dropout*, *weight decay*, *layer normalization*, etc. These techniques enhance the performance, generalization, and robustness of neural networks, making them essential components in the success of models across diverse domains, including computer vision and natural language processing.

The primary motivation behind regularization is to prevent model overfitting given a finite amount of available data. In online RL, the issue of overfitting does not occur explicitly, resulting in a limited body of work focused on this area, with most investigations centering on value function estimators (Kumar et al., 2020b; Kostrikov et al., 2021; An et al., 2021). In practice, applying regularization techniques to online RL often does not yield significant benefits. However, in the offline RL setting, the goal is to learn an optimal policy using a fixed dataset collected by other agents. This distinction makes offline RL conceptually more aligned with the use of regularization techniques. Furthermore, a recent study by Park et al. (2024) highlights that the actor networks are a critical bottleneck for the performance of offline RL algorithms. We hypothesize that common DL regularization techniques may alleviate this issue.

Motivated by the above considerations, this work aims to evaluate whether DL regularization techniques offer benefits when applied to actor networks in the continuous deep offline RL setting. Drawing inspiration from Park et al. (2024) and Tarasov et al. (2024c), we select the D4RL benchmark (Fu et al., 2020) and utilize the ReBRAC (Tarasov et al., 2023) and IQL (Kostrikov et al., 2021) algorithms for our experiments.

In this work, we aim to answer the following questions:

- Are DL regularization techniques beneficial for offline RL? By answering this question, we reduce the existing gap in the application of various regularizations to RL.

- How can different regularization techniques be effectively combined? It is known that in RL, a combination of multiple tricks might lead to further performance boost (Hessel et al., 2018; 2021; Tarasov et al., 2023).

- Do regularizations improve policy generalization? Actor generalization was claimed to be one of the core bottlenecks that should be eliminated in offline RL (Park et al., 2024). Regularizations may potentially help with this issue.

- How sensitive are different techniques to their hyperparameter choices? RL is known for being extremely sensitive to the choice of hyperparameters (Eimer et al., 2023), so we address this question in our study.

- What changes occur within actor networks when regularizations are applied? We wish to understand whether the performance improvements can be explained by interpretable changes inside networks.

## 2 Preliminaries

### 2.1 Offline Reinforcement Learning

We follow the conventional definition of RL problems as Markov Decision Processes, represented by the tuple $(S, A, P, R, \gamma)$, where: $S \subset \mathbb{R}^n$ denotes the state space, $A \subset \mathbb{R}^m$ denotes the action space, $P : S \times A \to S$ is the transition function, $R : S \times A \to \mathbb{R}$ is the reward function, and $\gamma \in (0, 1)$ is the discount factor. The objective of the RL agent is to maximize the expected sum of discounted rewards: $\sum_{t=0}^{\infty} \gamma^t R(s_t, a_t)$.

In online RL, the agent learns through trial and error by interacting with the environment. In contrast, the offline RL setup differs in that the agent does not interact with the environment in any way and must instead learn from a static dataset of transitions $D$ collected by other agents. This setup introduces additional challenges, such as the need to handle out-of-distribution states and actions.

### 2.2 Deep Learning regularization techniques

Regularization techniques are essential in deep learning to prevent overfitting and improve model generalization. In this section, we provide an overview of several regularization methods that are used in our work.

**Dropout (DO) (Hinton et al., 2012).** Dropout is a widely used regularization technique that mitigates overfitting by randomly setting a fraction of units in a neural network layer to zero during training. This prevents the network from becoming too reliant on specific neurons and forces it to learn more robust features. During inference, all neurons are used, but their output is scaled by the dropout rate to maintain consistency with the training phase.

**Weight Decay (Goodfellow, 2016).** Weight decay is a regularization technique that penalizes large weights in a neural network by adding a regularization term to the loss function. This term is often a combination of $L_1$ and $L_2$ norms, which is also known as the *elastic net* regularization. The loss function with elastic net regularization is given by:
$$\mathcal{L}(\theta) = \mathcal{L}_{original}(\theta) + \omega \left( \alpha ||\theta||_1 + (1 - \alpha)||\theta||_2^2 \right)$$
where $\mathcal{L}_{\text{original}}(\theta)$ is the original loss function, $|\theta|_1$ is the $L_1$ norm, and $|\theta|_2^2$ is the $L_2$ norm, and $\omega$ controls the weight of the penalty. The coefficient $\alpha \in (0, 1)$ controls the dominance between the $L_1$ and $L_2$ penalties. In our experiments, we call regularization **L1** when $\alpha = 1$, **L2** when $\alpha = 0$, and **elastic net (EN)** when $\alpha = 0.5$.

**Normalization Approaches (LN).** Normalization approaches aim to stabilize and accelerate training by normalizing input to each layer. We consider the following:

- **LayerNorm (Ba et al., 2016):** Normalizes across all features in a layer for each data point, ensuring that the mean and variance are consistent across features.

- **FeatureNorm (Yang et al., 2020):** Normalizes across individual features, often used in convolutional layers where each feature map is normalized independently.

- **GroupNorm (Wu & He, 2018):** Divides channels into groups and normalizes each group independently, providing a balance between instance normalization and batch normalization.

- **SpectralNorm (Miyato et al., 2018):** Normalizes the spectral norm of the weight matrices, controlling the Lipschitz constant of the network, and stabilizing training.

We chose LayerNorm, FeatureNorm and GroupNorm due to their implementation simplicity and the fact that they are all similar to LayerNorm which was shown to be a very useful modification in offline RL (Tarasov et al., 2023). The choice of SpectralNorm is motivated by its successful application in another RL domain (Gogianu et al., 2021).

**Input Noise (InN) (An, 1996).** Input noise involves adding small, random perturbations to the input data during training. This technique can be seen as a form of data augmentation that improves the robustness of the model by forcing it to learn features that are invariant to small input changes. It can also prevent the model from becoming too sensitive to specific input patterns.

**Objective Noise (BCN) (Wang & Principe, 1999; Müller et al., 2019).** Objective noise is a regularization technique where noise is added to the target values used in the loss function during training. This approach involves perturbing the target outputs slightly before computing the loss, which can help in preventing the model from overfitting to specific target values and improve its generalization capabilities.

**Gradient Noise (GrN) (Neelakantan et al., 2015).** Gradient noise is a regularization technique where noise is added to the gradients during the optimization process. This approach can help the model escape local minima and saddle points in the loss landscape, potentially leading to better generalization. By injecting noise into the gradient updates, the model may explore a broader range of solutions, reducing the risk of overfitting.

## 3 Methodology

### 3.1 General experimental design

**Benchmark.** For our experiments, we utilize the popular D4RL (Fu et al., 2020) benchmark as our testbed. Following the selection of tasks in Park et al. (2024), we focus on the medium datasets for Gym-MuJoCo tasks, medium- and large-diverse datasets for AntMaze tasks, and cloned datasets for Adroit. In sections subsection 4.1, subsection 4.2, and subsection 4.5, we reserve 5% of the data as a validation set to observe the differences in tracked metrics, aiming to determine whether regularizations offer benefits in offline RL and to understand their impact on the actor network's internal structure. For the results presented in subsection 4.3 and subsection 4.4, we use complete data sets for training, evaluating the entire set of Gym-MuJoCo, AntMaze, and Adroit tasks. These sections focus on assessing the extent to which regularizations can enhance performance and improve the generalization of actor networks.

**Algorithms.** To enable a study with sufficient evidence, we select two competitive, ensemble-free algorithms: ReBRAC (Tarasov et al., 2023), a modification of minimalist TD3+BC (Fujimoto & Gu, 2021) policy-regularized approach, and IQL (Kostrikov et al., 2021), a powerful algorithm from the implicit regularization family. We use the ReBRAC + CE + MT modification of Tarasov et al. (2024a), where ReBRAC is augmented with a cross-entropy objective (Farebrother et al., 2024), leading to significant performance improvements and improved training stability on various tasks. For brevity, this modification is simply referred to as ReBRAC until specified. The choice of these two algorithms is motivated by the findings in Park et al. (2024) and Tarasov et al. (2024c), where the policy-regularized methods, such as ReBRAC, were identified as the most promising for offline RL, and where IQL was recognized as a strong ensemble-free alternative. When modifying ReBRAC, we implement large batch optimization (Nikulin et al., 2022) for Adroit tasks, ensuring that this aspect of ReBRAC is consistent in all tasks. It is important to note that the original ReBRAC and it's ReBRAC+CE upgrade did not maintain this level of hyperparameter consistency; in subsection 4.3 and subsection 4.4 we compare modifications against ReBRAC+CE with large batch on Adroit. Additionally, for the IQL baseline, we remove *dropout* for Adroit tasks to better isolate and examine the impact of the regularization techniques. Due to computational constraints and ReBRAC's performance, we primarily focus our experiments on it in subsection 4.2. More details on algorithms are provided in Appendix B.

**Evaluation metrics.** Given the inherent noise in RL training processes, we use **Running Average Returns (RAR)** (Kang et al., 2023) over the last $N$ checkpoints for sections subsection 4.1 and subsection 4.2. Here, $N = 5$ for AntMaze tasks[1] and $N = 10$ for other tasks. In the main body of our work, we present our results using *rliable* metrics (Agarwal et al., 2021), with some raw scores provided in the Appendix for reference. Inspired by Moalla et al. (2024), for investigating the internal effects of regularizations in section subsection 4.5, we calculated the following metrics using both training and validation data for the penultimate actor layer: fraction of dead neurons (in terms of the ReLU activation used), feature norms, the feature PCA rank approximation (Kumar et al., 2020a) and plasticity loss (Lyle et al., 2022)[2].

## 3.2 Regularizations leveraging

We provide pseudocodes for the regularizations usages in Appendix C.

**Dropout and Normalizations.** In our experiments we place dropout and normalization layers after the activation function of each hidden layer within the actor network. When both regularizations are applied simultaneously, dropout is applied after the normalization, which is a conventional approach. Note, that we do not combine normalizations with each other and apply only one if any.

**Noise regularization** For each noise-based regularization, we modify the relevant values – such as inputs, objective targets, or gradients—using the following approach:

$$\tilde{y} = y + \nu\epsilon$$

where $y$ represents the original value, $\epsilon \sim \mathcal{N}(0, 1)$ is drawn from a standard normal distribution, and $\nu$ is a hyperparameter controlling the noise magnitude.

Following Neelakantan et al. (2015), we apply a decayed version of the gradient noise. Gradient noise decays over time according to $\frac{\nu_{gr}}{(1+t)^\gamma}$, where $t$ is the number of training iterations, and $\gamma$ is set to the recommended value of 0.55. We found this decayed approach to be more effective than using a constant noise level throughout training.

For objective noise, the modifications differ based on the algorithm:

- **ReBRAC**: The objective noise modifies the actor's behavior cloning (BC) term as follows:

$$\mathcal{L}_{BC}(\theta) = (a_\theta - a + \nu_{ob}\epsilon)^2$$

  where $a_\theta$ is the action predicted by the actor, $a$ is the action from the dataset, and $\nu_{ob}$ controls the noise level.

- **IQL**: For IQL, objective noise is introduced into the policy extraction process:

$$\mathcal{L}_{AWR}(\theta) = \exp[\beta(Q(s, a) - V(s)) + \nu_{ob}\epsilon] \log \pi_\theta(a|s)$$

  where $(s, a)$ is a state-action pair from the dataset, $Q(s, a)$ is the action-value function, $V(s)$ is the value function, and $\beta$ is a temperature parameter. The noise term $\nu_{ob}\epsilon$ is added to the advantage function.

# 4 Experimental Results

## 4.1 Which regularizations independently help actors?

In our initial experiments, we investigate the impact of individually applying each regularization technique on the performance of the algorithms. Using the D4RL subset described in section 3, we report the rliable RAR metrics in Figure 1.

---

[1]Following previous studies, AntMaze tasks are evaluated over more episodes, and to keep compute time tractable, the frequency of evaluations is reduced.

[2]We provide further details in Appendix A

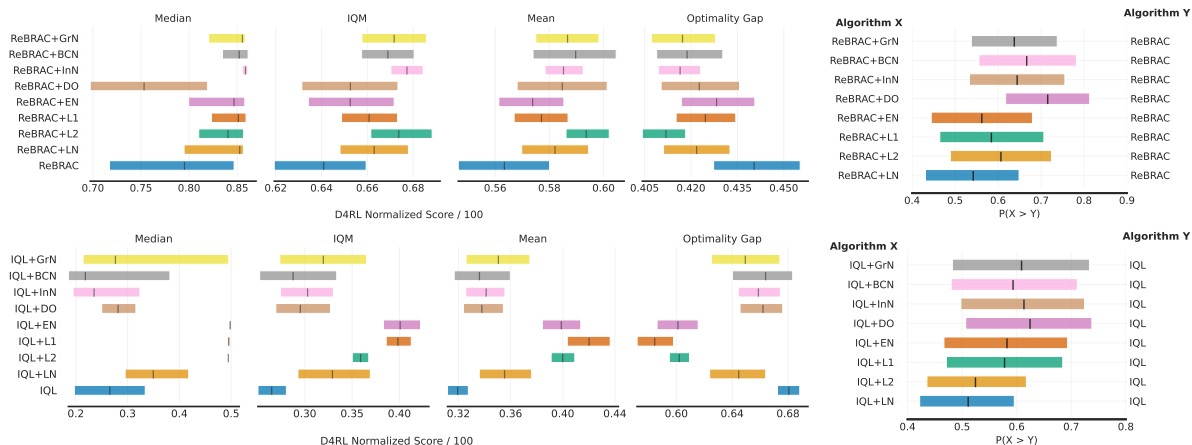

Figure 1: rliable metrics for RAR scores averaged over D4RL subset when regularizations are applied individually. Top row ReBRAC performance, bottom row IQL performance. Left graphs Median, IQM, Mean and Optimality Gap, right graphs Probability of Improvement against original algorithm. Each task was run over 5 random seeds.

The results indicate that adding any regularization with an appropriate hyperparameter setting to ReBRAC and IQL generally enhances algorithm performance. For ReBRAC, L2 regularization and various noise-based techniques provide the most significant improvements, while for IQL, weight decays emerge as the most promising modifications.

In Appendix E, we present these metrics in relation to the choice of hyperparameters for each regularization technique, allowing us to draw several key conclusions. LayerNorm proves to be the most effective Layer Normalization technique; it outperforms other normalization methods in IQL and matches their performance in ReBRAC. ReBRAC is relatively insensitive to the choice of the weight decay hyperparameter $\omega$, but its performance degrades significantly when the penalty is too large. In contrast, IQL benefits from higher values of $\omega$.

Interestingly, dropout only improves performance at low dropout rates below 0.3, with 0.1 being the optimal choice among the options we tested. This aligns with the original IQL settings[3]. ReBRAC is highly sensitive to the magnitude of input and gradient noise, with lower values proving more effective, while IQL is more robust to these choices and performs well with intermediate values. For both ReBRAC and IQL, objective noise is beneficial across a wide range of noise levels.

## 4.2 Can we combine regularizations?

In this set of experiments, we explore whether combining multiple regularization techniques can lead to better performance compared to using them individually. We present the RAR rliable metrics, including the median, interquartile mean (IQM), mean, and optimality gap scores in Figure 2, with probabilities of improvement detailed in Appendix F. We use the following notation for the modifications: *Algorithm+MODs+TUNED*, where *MODs* are modifications with already tuned hyperparameters when they were applied without *TUNED* and *TUNED* is the addition to the modifications prefix who's parameters are tuned. This experimental setup allows us to identify whether regularizations run into conflicts with each other and require additional hyperparameters tuning when combined.

The results for ReBRAC show that combining regularizations generally improves performance in most cases. Notably, the combination of dropout (with a rate of 0.1) and LayerNorm appears to be a strong default choice, as it integrates well with other techniques. However, weight decays are less compatible with other regularizations. The best combination we identified for ReBRAC was dropout + LayerNorm + gradient noise

---

[3]It is worth noting that the IQL authors do not provide an analysis or commentary on the role of dropout.

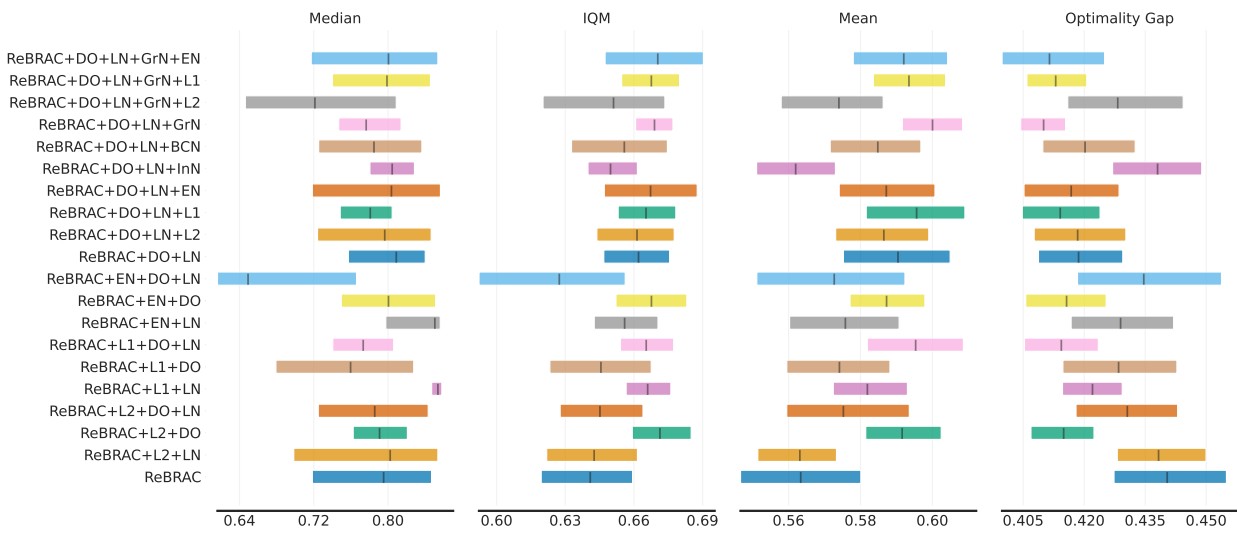

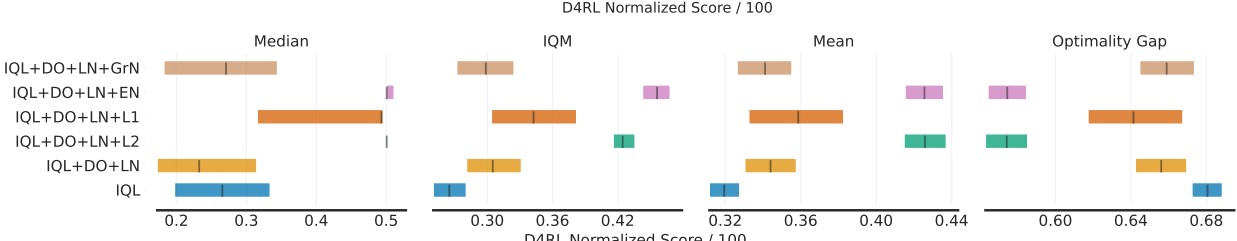

Figure 2: Median, IQM, Mean and Optimality Gap rliable metrics for RAR scores averaged over D4RL subset when regularizations are combined. Top row ReBRAC performance, bottom row IQL performance. Each task was run over 5 random seeds.

(DO+LN+GrN). Unfortunately, the results did not reveal a consistent pattern across different combinations, making it difficult to predict the outcome based solely on the individual performance of each regularization.

One clear takeaway is that hyperparameters that work well individually may not perform optimally when combined. For example, the combination of elastic net, dropout, and LayerNorm (EN+DO+LN) performed differently when the order of tuning was changed to DO+LN+EN, highlighting the importance of tuning when combining regularizations.

Based on our findings from ReBRAC and the performance of individual regularizations, we conducted a smaller set of experiments with IQL. We tested the combination of dropout and LayerNorm (DO+LN) because it consistently showed promise, as well as its combinations with weight decay techniques, which performed the best individually for IQL. We also tested DO+LN+GrN, as this combination yielded the best results for ReBRAC. The results indicate that DO+LN+EN was the most effective combination for IQL. Interestingly, while L1 regularization was the best individual modification, the combination of DO+LN+L1 was outperformed by L2 and elastic net (EN) when used together.

### 4.3 Per-dataset tuning

In the previous subsections, we established that incorporating deep learning regularization techniques into the actor network generally enhances algorithm performance. However, the results reported were based on the best hyperparameter settings across various tasks, which may not be optimal for individual datasets. For instance, we observed that dropout, even at low rates, can decrease performance on the antmaze-large-diverse

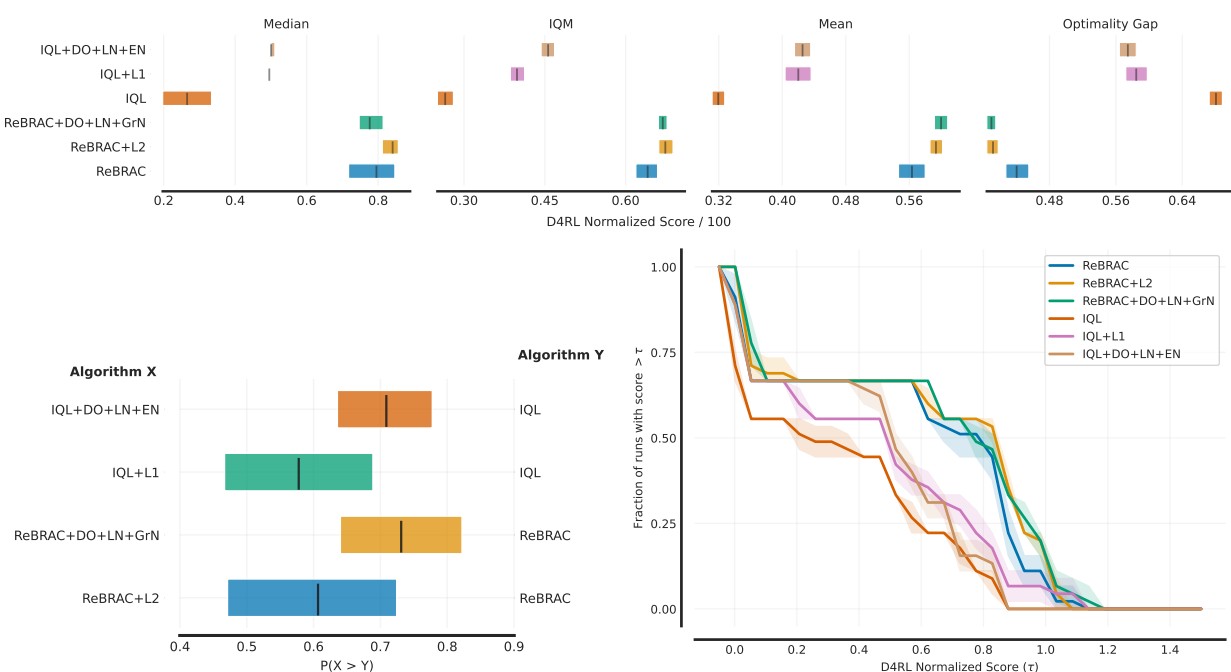

Figure 3: rliable metrics for last checkpoint performance averaged over all Gym-MuJoCo, AntMaze, and Adroit datasets when regularizations hyperparameters are tuned per dataset. Top row Median, IQM, Mean and Optimality Gap, bottom Probability of Improvement, and Performance Profiles. Every task evaluated over 10 seeds.

Table 1: Final checkpoint mean performance averaged over D4RL domains. Every task evaluated over 10 seeds.

|  | **ReBRAC** | | | **IQL** | | |
|---|---|---|---|---|---|---|
| **Domain** | Original | L2 | DO+LN+GrN | Original | L1 | DO+LN+EN |
| Gym-MuJoCo | 81.5 | 80.8 | **81.7** | 74.6 | 73.8 | **74.8** |
| AntMaze | 90.3 | 90.4 | **92.6** | 61.1 | 62.1 | **66.0** |
| Adroit w\o expert | 19.2 | 22.1 | **23.8** | 4.2 | **26.7** | 9.8 |
| Adroit | 53.2 | 55.6 | **57.2** | 37.5 | **58.4** | 47.3 |
| Avg | 73.5 | 74.0 | **75.4** | 60.0 | **66.7** | 64.2 |

dataset, whereas cloned Adroit tasks benefit from higher dropout rates. Many recent offline RL studies present results where hyperparameters are fine-tuned specifically for each dataset, an approach that maximizes algorithm performance and aligns with real-world scenarios where practitioners are likely to tailor parameters to particular tasks.

To explore the effects of per-dataset tuning, we selected the best-performing individual and combined regularizations from previous sections: L2 and DO + LN + GrN for ReBRAC, and L1 and DO + LN + EN for IQL.

Following the evaluation protocol outlined by Tarasov et al. (2023), we tuned hyperparameters using a different set of random seeds than those used for final evaluation. This approach ensures a fair comparison by preventing overfitting to specific random seeds or initializations.

We report the final checkpoint performance in Figure 3 and Table 1. The results clearly demonstrate that regularizations consistently benefit algorithm performance on sparse tasks (AntMaze) and tasks with a very high-dimensional action spaces and limited state-action space coverage (Adroit). Adroit results demonstrate

that regularizations may help to reduce the overfitting problem when it occurs in offline RL. At the same time, the performance remains the same on the simpler tasks (Gym-MuJoCo). While combined regularizations tend to yield more substantial improvements, in some cases, a single modification may offer greater gains. Notably, we achieved a state-of-the-art score of 92.7 on AntMaze tasks, outperforming all known offline RL approaches, using the enhanced minimalist approach ReBRAC. Per-dataset results and comparisons with other algorithms are detailed in Appendix G.

## 4.4 Actor generalization

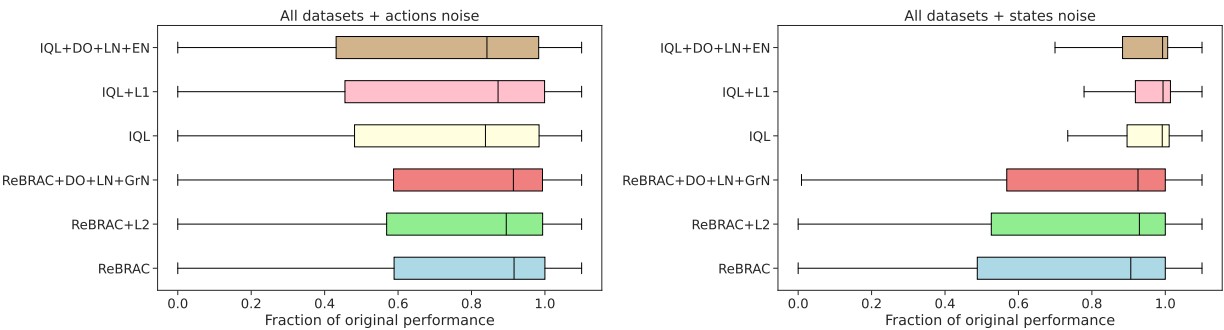

Figure 4: Final checkpoint performance with noise injection divided by the noise-free performance. We clip outliers at the level of 1.1 due to the metric sensitivity to some Adroit tasks.

Next, we aimed to determine whether regularizations enhance the generalization abilities of the algorithms. To assess this, we introduced Gaussian noise during inference to either the actions produced by the actor ($\sigma = 0.2$) or the observations ($\sigma = 0.05$). Adding noise to the actions simulates the impact of deploying the policy in a more stochastic environment, while noise in the states evaluates the actor's robustness to input perturbations. The results of these experiments are presented in Figure 4, with per-domain results available in Appendix H.

Our findings indicate that the proposed regularization techniques do not lead to significant generalization improvements for ReBRAC or IQL in either scenario. Interestingly, on average ReBRAC exhibits greater robustness to action perturbations, whereas IQL demonstrates more resilience to noisy states.

## 4.5 What is happening inside actor?

In this section, we explore the internal changes within the actor network when various regularization techniques are applied. We monitor several metrics by comparing them between training and validation sets, focusing on the output of the penultimate linear activation layer. These differences are of particular interest as they reveal how the actor perceives potential out-of-distribution data. The results are illustrated in Figure 5. We also track the network plasticity by optimizing Behaviour Clonning term using the validation set. Per-domain results are shown in Appendix I.

For both ReBRAC and IQL, applying normalization results in a complete elimination of dead neurons, which is generally considered a desirable property. This adjustment makes the validation data nearly indistinguishable from the training data in terms of feature norms for ReBRAC, but not for IQL. Weight decay also reduces the number of dead neurons in the validation set for both algorithms and has varied effects on other metrics altering the internal representations. Dropout, on the other hand, primarily reduces the number of dead neurons without significantly affecting other metrics. The introduction of noise does not significantly impact the tracked metrics for either algorithm. Finally, plasticity changes depend on the considered algorithm-domain pair. Layer normalization improves ReBRAC's plasticity on AntMaze tasks while dropout notably decreases it which aligns with performance drop when dropout applied to some of the AntMaze tasks. All of the regularizations improve IQL plasticity on Adroit domain. In other cases we do not observe significant changes.

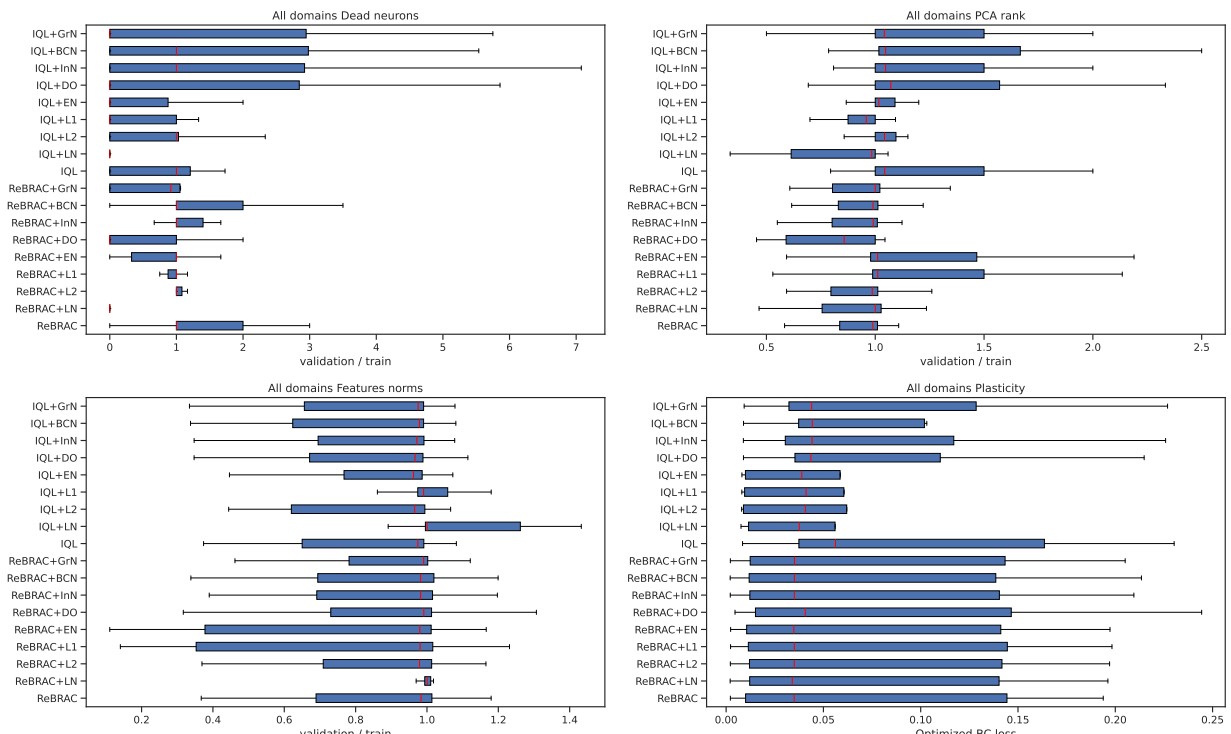

Figure 5: Validation data metrics divided by the train data metrics from actor penultimate linear layer of the final checkpoint averaged over subset of D4RL datasets.

Our findings demonstrate that regularizations improve performance by somehow imporing the internal representations. Amount of dead neurons is affected the most by regularizations, however layer normalization (which completely eliminates them) results from subsection 4.1 demonstrate that this is not the core problem.

## 5    Related Work

Offline Reinforcement Learning (RL) (Levine et al., 2020) is a rapidly advancing field with numerous approaches being introduced to address the challenges of out-of-distribution state-action pairs and policy learning from static datasets. Prominent methods such as CQL (Kumar et al., 2020b), IQL (Kostrikov et al., 2021), TD3+BC (Fujimoto & Gu, 2021), and others (An et al., 2021; Chen et al., 2021; Akimov et al., 2022; Yang et al., 2022; Ghasemipour et al., 2022; Nikulin et al., 2023) have progressively tackled these challenges with increasingly sophisticated solutions. However, as these methods have evolved, they have also become more nuanced and complex.

ReBRAC (Tarasov et al., 2023) demonstrated that augmenting a minimalist offline RL approach TD3+BC (Fujimoto & Gu, 2021) with straightforward design choices can yield state-of-the-art performance across a variety of tasks. Notably, most of ReBRAC's enhancements focused on the critic component of the actor-critic architecture. Inspired by this work and the findings of Park et al. (2024), which highlighted the actor network as a potential bottleneck in offline RL, our research explores the impact of applying popular DL regularization techniques specifically to actor networks, aiming to unlock their potential benefits.

DL regularization techniques have been extensively studied in the context of supervised learning (Kukačka et al., 2017; Smith, 2018; Tian & Zhang, 2022), where their use has become standard practice in the development of robust models. However, their application to RL has been limited. For instance, Hiraoka et al. (2021) explored the application of dropout and LayerNorm in RL, showing performance improvements, and Smith et al. (2022) found that such modifications were critical in real-world tasks. Additionally, Bhatt et al. (2019) revealed that the application of BatchNorm in RL is nuanced, with performance gains not easily

achievable through straightforward implementation. In the context of offline RL, a few studies have shown that applying layer normalizations to the critic network can be beneficial (Kumar et al., 2022; Nikulin et al., 2022; Ball et al., 2023; Tarasov et al., 2023). However, all mentioned works focus on critic modifications and largely restrict their exploration to dropout and LayerNorm. To our knowledge, IQL (Kostrikov et al., 2021) is the only notable method where dropout is applied to the actor network in some tasks, yet no analysis of this application has been provided.

Our work builds on these insights by systematically evaluating the impact of various DL regularizations on actor networks within offline RL, filling a gap in the existing literature and contributing to the broader understanding of how these techniques can enhance RL performance.

## 6  Conclusion

This work demonstrates the clear benefits of applying DL regularization techniques to actor networks within the offline RL setup when evaluation budget allows some hyperparameters search. By systematically evaluating the impact of various regularizations, we have not only shown performance improvements across multiple tasks but also provided insights into the internal mechanisms driving these gains. Our findings help to bridge the gap between the extensive use of DL regularizations in supervised learning and their limited application in RL, particularly in the context of actor networks.

While our study focuses on two algorithm ReBRAC and IQL – the techniques we explored are broadly applicable and not inherently tied to any specific algorithm. This suggests that the benefits observed here could extend to other RL frameworks and architectures, opening avenues for future research.

Despite the promising results, our work has some limitations. We only considered a specific subset of regularizations and applied them in an offline RL context. Future research could explore a wider range of regularization methods, including those tailored for different network architectures or more complex tasks. Additionally, extending these techniques to other RL setups, such as offline-to-online or fully online scenarios, could yield further insights and performance improvements.

Moreover, investigating the scalability of these regularizations in large-scale, real-world environments and their interaction with other RL components, such as critics and exploration strategies, could provide a more comprehensive understanding of their potential. By deepening our understanding of how DL regularizations influence actor network behavior, we move closer to more robust and generalizable RL algorithms, capable of tackling increasingly complex tasks with greater efficiency and reliability.

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

# A Experimental details

For results in subsection 4.3, we conducted a hyperparameter search and selected the best results using RAR for each dataset. We used JAX implementations of ReBRAC+CE and IQL from Tarasov et al. (2024a) based on the Clean Offline RL (Tarasov et al., 2024c) ReBRAC implementation. Adam Kingma & Ba (2014) optimizer is used for both algorithms

The experiments were conducted on Titan RTX GPUs.

Our study utilized the v2 version of datasets for Gym-MuJoCo and AntMaze, and v1 for Adroit. The agents were trained for one million steps in all domains and evaluated over ten episodes for Gym-MuJoCo and Adroit and over one hundred episodes for AntMaze. Following Chen et al. (2022), AntMaze reward function is multiplied by 100.

For weight decays we tuned $\omega$ parameter from the set $\{0.00001, 0.0001, 0.001, 0.01, 0.1\}$, for dropout we tuned values from $\{0.1, 0.2, 0.3, 0.5, 0.75, 0.9\}$ for ReBRAC and $\{0.1, 0.2, 0.3, 0.5\}$ for IQL, and noises parameters $\nu_{InN}, \nu_{BCN}, \nu_{GrN}$ from $\{0.003, 0.01, 0.03, 0.1, 0.3\}$.

When measuring the actor plasticity loss for ReBRAC and IQL we use validation data with Behaviour Cloning (BC) term (MSE loss) running over 100 update steps. The final loss value is a metric that we use in subsection 4.5. This approach allows us to obtain scalar values which demonstrate the ability of the trained policy to update it's predictions using the new data.

# B Algorithms overview

Here, we provide a short description of the used algorithms.

**Revisited BRAC (ReBRAC).** ReBRAC (Tarasov et al., 2024b) stands as a minimalist, state-of-the-art ensemble-free algorithm for both offline and offline-to-online RL. It employs MSE to penalize deviations from actions present in the dataset. Built upon TD3+BC (Fujimoto & Gu, 2021), ReBRAC incorporates several modifications that significantly enhance its performance. The Q-loss function takes the form:

$$\mathbb{E}_{(s,a,s',\hat{a}')\sim D, a'\sim \pi(s')}(Q_\theta(s,a) - [R(s,a) + \gamma(Q_{\hat{\theta}}(s',a') - \beta_1(a' - \hat{a}')^2)])^2$$

The policy loss function is:

$$\mathbb{E}_{(s,a)\sim D, a'\sim \pi(s')}[-Q_\theta(s,a') + \beta_2(a' - a)^2)]$$

$\beta_1$ and $\beta_2$ are regularization hyperparameters.

We select ReBRAC as an exemplary algorithm with policy regularization (i.e. forcing the policy to commit actions simillar to dataset) due to its high performance and simplicity.

**Implicit Q-Learning (IQL).** IQL (Kostrikov et al., 2021) represents another competitive offline and offline-to-online RL algorithm. Its key advantage lies in its exclusion of out-of-distribution examples during training, a departure from most other offline RL approaches. This is achieved through the training of the V-function with the following loss:

$$\mathbb{E}_{(s,a)\sim D} L_2^\tau(Q_\theta(s,a) - V_\phi(s))$$

where $L_2^\tau(u) = |\tau - \mathbb{I}(u < 0)|u^2$ is an expectile loss function with parameter $\tau$.

The Q-function loss takes the form:

$$\mathbb{E}_{(s,a,s')\sim D}(Q_\theta(s,a) - [R(s,a) + \gamma V_\phi(s')])^2$$

And the policy loss is:

$$\mathbb{E}_{(s,a)\sim D} \exp[\beta(Q_\theta(s,a) - V_\phi(s))] \log \pi(a|s)$$

where $\beta$ is algorithm hyperparameter.

Regularization is achieved through a specially designed V-function loss, without explicit penalties for the policy or value functions. We opt for IQL as it is the best example of algorithms within this family.

## C    Pseudocodes

In this section we provide JAX-based partial code snippets used in our experiments. Some details are ommited, see our source code for more details.

Listing 1: Actors forward function with Layer Norms and Dropout in Python JAX.

```python
@nn.compact
def forwrad(self, state: jax.Array):
    layers = [
        nn.Dense(self.hidden_dim),
        nn.relu,
        nn.LayerNorm() if self.ln else id,
        nn.LayerNorm(use_bias=False, use_scale=False) if self.fn else id,
        nn.GroupNorm() if self.gn else id,
        nn.Dropout(rate=self.dropout),
    ]
    for _ in range(self.n_hiddens - 2):
        layers += [
            nn.Dense(self.hidden_dim),
            nn.relu,
            nn.LayerNorm() if self.ln else id,
            nn.LayerNorm(use_bias=False, use_scale=False) if self.fn else id,
            nn.GroupNorm() if self.gn else id,
            nn.Dropout(rate=self.dropout),
        ]

    net = nn.Sequential(layers)

    # Spectral norm is added only after penultimate layer if needed
    trunk = nn.Dense(self.hidden_dim,)(net(state)) \
        if not self.sn else (
        nn.SpectralNorm(nn.Dense(
            self.hidden_dim,
        ))(net(state), update_stats=train))

    last_layer = nn.Sequential(
        [
            nn.relu,
            nn.LayerNorm() if self.ln else id,
            nn.LayerNorm(use_bias=False, use_scale=False) if self.fn else id,
            nn.GroupNorm() if self.gn else id,
            nn.Dropout(rate=self.dropout),
            nn.Dense(self.action_dim),
            nn.tanh
        ]
    )
    actions = last_layer(trunk)

    return actions, trunk
```

Listing 2: General actor update in Python JAX.

```python
def update_actor(
        key: jax.random.PRNGKey, actor: TrainState, critic: TrainState,
        batch: Dict[str, jax.Array], input_noise: float, bc_noise: float, grad_noise: float,
):
    key, input_noise_key, bc_noise_key, grad_noise_key = jax.random.split(key, 4)

    in_noise = jax.random.normal(input_noise_key, batch["states"].shape) * input_noise
    b_noise = jax.random.normal(bc_noise_key, batch["actions"].shape) * bc_noise

    def actor_loss_fn(params: jax.Array):
        actor_outputs = actor.apply_fn(params, batch["states"] + in_noise)
        # algorithm specific loss function with b_noise added
        loss = ...
        return loss

    grads = jax.grad(actor_loss_fn)(actor.params)

    def add_gaussian_noise(gr, noise_std, rng_key):
        def add_noise_to_grad(g, rng_key):
            noise = jax.random.normal(rng_key, g.shape) * noise_std / ((1 + actor.step) **
                0.55)
            return g + noise

        leaves, tree = jax.tree_util.tree_flatten(gr)
        rng_keys = jax.random.split(rng_key, num=len(leaves))
        rng_keys = jax.tree_util.tree_unflatten(tree, rng_keys)

        noisy_grads = jax.tree_util.tree_map(lambda g, k: add_noise_to_grad(g, k), gr,
            rng_keys)
        return noisy_grads

    grads = add_gaussian_noise(grads, grad_noise, grad_noise_key)
    new_actor = actor.apply_gradients(grads=grads)

    return key, new_actor
```

Listing 3: Full code of our custom implementation of AdamW to support elastic net penalty. Note, it might be incompatible with newer JAX/Optax versions.

```
1  AddDecayedWeightsState = base.EmptyState
2
3  def add_elastic_weights(
4      weight_decay: Union[float, jax.Array] = 0.0,
5      l1_ratio: float = 0.0,
6      mask: Optional[Union[Any, Callable[[base.Params], Any]]] = None
7  ) -> base.GradientTransformation:
8    def init_fn(params):
9      del params
10     return AddDecayedWeightsState()
11
12   def update_fn(updates, state, params):
13     if params is None:
14       raise ValueError(base.NO_PARAMS_MSG)
15     updates = jax.tree_util.tree_map(
16         lambda g, p: g + weight_decay * ((1 - l1_ratio) * p + l1_ratio * jnp.sign(p)),
               updates, params)
17     return updates, state
18
19   if mask is not None:
20     return wrappers.masked(
21         base.GradientTransformation(init_fn, update_fn), mask)
22   return base.GradientTransformation(init_fn, update_fn)
23
24
25  def adamw_elastic(
26         learning_rate: base.ScalarOrSchedule,
27         b1: float = 0.9,
28         b2: float = 0.999,
29         eps: float = 1e-8,
30         eps_root: float = 0.0,
31         mu_dtype: Optional[Any] = None,
32         weight_decay: float = 1e-4,
33         l1_ratio: float = 0.0,
34         mask: Optional[Union[Any, Callable[[base.Params], Any]]] = None,
35         *,
36         nesterov: bool = False,
37  ) -> base.GradientTransformation:
38
39      return combine.chain(
40          transform.scale_by_adam(
41              b1=b1,
42              b2=b2,
43              eps=eps,
44              eps_root=eps_root,
45              mu_dtype=mu_dtype,
46              nesterov=nesterov,
47          ),
48          add_elastic_weights(weight_decay, l1_ratio, mask),
49          transform.scale_by_learning_rate(learning_rate),
50      )
51
52  # Optax usage
53  optimizer = adamw_elastic(
54      learning_rate=config.lr, weight_decay=config.wd, l1_ratio=config.l1_ratio
55  )
```

# D   Hyperparameters

## D.1   ReBRAC+CE

Table 2: ReBRAC+CE general hyperparameters.

| Parameter | Value |
|---|---|
| batch size | 1024 |
| learning rate (all networks) | 1e-3 |
| tau ($\tau$) | 5e-3 |
| hidden dim (all networks) | 256 |
| num hidden layers (all networks) | 3 |
| gamma ($\gamma$) | 0.999 on AntMaze, 0.99 otherwise |
| nonlinearity | ReLU |

Table 3: ReBRAC+CE regularizations best hyperparameters.

| Task Name | $\omega$ (L2) | Dropout rate (DO+LN+GrN) | Actor LN (DO+LN+GrN) | $\nu_{GrN}$ (DO+LN+GrN) |
|---|---|---|---|---|
| halfcheetah-random | 0.1 | 0.2 | false | 0.01 |
| halfcheetah-medium | 0.1 | 0.1 | true | 0.01 |
| halfcheetah-expert | 0.00001 | 0.0 | true | 0.003 |
| halfcheetah-medium-expert | 0.00001 | 0.0 | true | 0.003 |
| halfcheetah-medium-replay | 0.00001 | 0.2 | true | 0.01 |
| halfcheetah-full-replay | 0.01 | 0.0 | true | 0.01 |
| hopper-random | 0.001 | 0.2 | false | 0.01 |
| hopper-medium | 0.01 | 0.2 | true | 0.01 |
| hopper-expert | 0.00001 | 0.2 | false | 0.003 |
| hopper-medium-expert | 0.0001 | 0.0 | false | 0.003 |
| hopper-medium-replay | 0.00001 | 0.0 | true | 0.003 |
| hopper-full-replay | 0.01 | 0.0 | false | 0.003 |
| walker2d-random | 0.0001 | 0.1 | true | 0.01 |
| walker2d-medium | 0.01 | 0.1 | false | 0.0 |
| walker2d-expert | 0.1 | 0.2 | false | 0.003 |
| walker2d-medium-expert | 0.1 | 0.2 | false | 0.01 |
| walker2d-medium-replay | 0.001 | 0.2 | false | 0.0 |
| walker2d-full-replay | 0.1 | 0.1 | true | 0.0 |
| antmaze-umaze | 0.00001 | 0.2 | false | 0.01 |
| antmaze-umaze-diverse | 0.00001 | 0.1 | true | 0.0 |
| antmaze-medium-play | 0.0001 | 0.2 | true | 0.003 |
| antmaze-medium-diverse | 0.1 | 0.1 | true | 0.0 |
| antmaze-large-play | 0.00001 | 0.0 | true | 0.0 |
| antmaze-large-diverse | 0.0001 | 0.0 | true | 0.003 |
| pen-human | 0.1 | 0.1 | true | 0.01 |
| pen-cloned | 0.00001 | 0.1 | false | 0.003 |
| pen-expert | 0.001 | 0.1 | true | 0.0 |
| door-human | 0.1 | 0.1 | true | 0.01 |
| door-cloned | 0.1 | 0.2 | false | 0.003 |
| door-expert | 0.01 | 0.1 | true | 0.0 |
| hammer-human | 0.1 | 0.2 | true | 0.0 |
| hammer-cloned | 0.001 | 0.2 | true | 0.01 |
| hammer-expert | 0.0001 | 0.1 | true | 0.003 |
| relocate-human | 0.1 | 0.2 | true | 0.01 |
| relocate-cloned | 0.01 | 0.2 | false | 0.01 |
| relocate-expert | 0.01 | 0.2 | true | 0.003 |

## D.2 IQL

Table 4: IQL general hyperparameters.

| Parameter | Value |
|---|---|
| batch size | 256 |
| learning rate (all networks) | 3e-4 |
| tau ($\tau$) | 5e-3 |
| hidden dim (all networks) | 256 |
| num hidden layers (all networks) | 2 |
| gamma ($\gamma$) | 0.99 |
| nonlinearity | ReLU |
| learning rate decay | Cosine |

Table 5: IQL regularizations best hyperparameters.

| Task Name | $\omega$ (L1) | Dropout rate (DO+LN+EN) | Actor LN (DO+LN+EN) | $\omega$ (DO+LN+EN) |
|---|---|---|---|---|
| halfcheetah-random | 0.001 | 0.0 | false | 0.01 |
| halfcheetah-medium | 0.0001 | 0.0 | true | 0.01 |
| halfcheetah-expert | 0.0001 | 0.0 | true | 0.0 |
| halfcheetah-medium-expert | 0.0001 | 0.0 | false | 0.001 |
| halfcheetah-medium-replay | 0.01 | 0.1 | true | 0.01 |
| halfcheetah-full-replay | 0.0001 | 0.0 | false | 0.001 |
| hopper-random | 0.01 | 0.1 | true | 0.001 |
| hopper-medium | 0.01 | 0.2 | false | 0.01 |
| hopper-expert | 0.001 | 0.0 | false | 0.0 |
| hopper-medium-expert | 0.00001 | 0.0 | false | 0.0 |
| hopper-medium-replay | 0.01 | 0.1 | true | 0.01 |
| hopper-full-replay | 0.0001 | 0.2 | true | 0.001 |
| walker2d-random | 0.0001 | 0.0 | false | 0.0 |
| walker2d-medium | 0.01 | 0.1 | false | 0.001 |
| walker2d-expert | 0.01 | 0.1 | false | 0.01 |
| walker2d-medium-expert | 0.0001 | 0.1 | false | 0.01 |
| walker2d-medium-replay | 0.0001 | 0.1 | true | 0.001 |
| walker2d-full-replay | 0.01 | 0.2 | true | 0.001 |
| antmaze-umaze | 0.0001 | 0.0 | true | 0.001 |
| antmaze-umaze-diverse | 0.00001 | 0.0 | true | 0.0 |
| antmaze-medium-play | 0.00001 | 0.1 | false | 0.01 |
| antmaze-medium-diverse | 0.001 | 0.0 | false | 0.0 |
| antmaze-large-play | 0.001 | 0.0 | true | 0.001 |
| antmaze-large-diverse | 0.001 | 0.1 | true | 0.01 |
| pen-human | 0.01 | 0.1 | false | 0.01 |
| pen-cloned | 0.1 | 0.1 | false | 0.01 |
| pen-expert | 0.1 | 0.0 | true | 0.001 |
| door-human | 0.01 | 0.0 | true | 0.001 |
| door-cloned | 0.1 | 0.2 | true | 0.001 |
| door-expert | 0.0001 | 0.2 | false | 0.01 |
| hammer-human | 0.001 | 0.0 | false | 0.01 |
| hammer-cloned | 0.01 | 0.0 | true | 0.01 |
| hammer-expert | 0.0001 | 0.1 | false | 0.0 |
| relocate-human | 0.001 | 0.0 | false | 0.0 |
| relocate-cloned | 0.001 | 0.2 | true | 0.0 |
| relocate-expert | 0.01 | 0.1 | true | 0.01 |

# E  Hyperparameters sensitivity

Sensitivity to the hyperparameters choice for all of the regularization techniques based on the experiments from subsection 4.1.

## E.1  Layer Normalizations

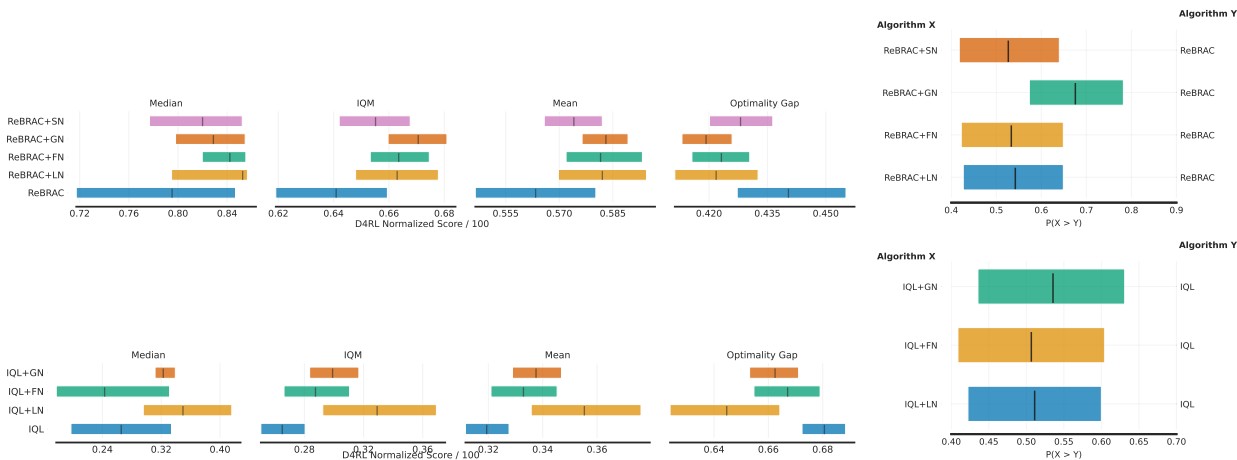

Figure 6: rliable metrics for RAR scores averaged over D4RL subset for different Layer Normalizations.

## E.2  L2

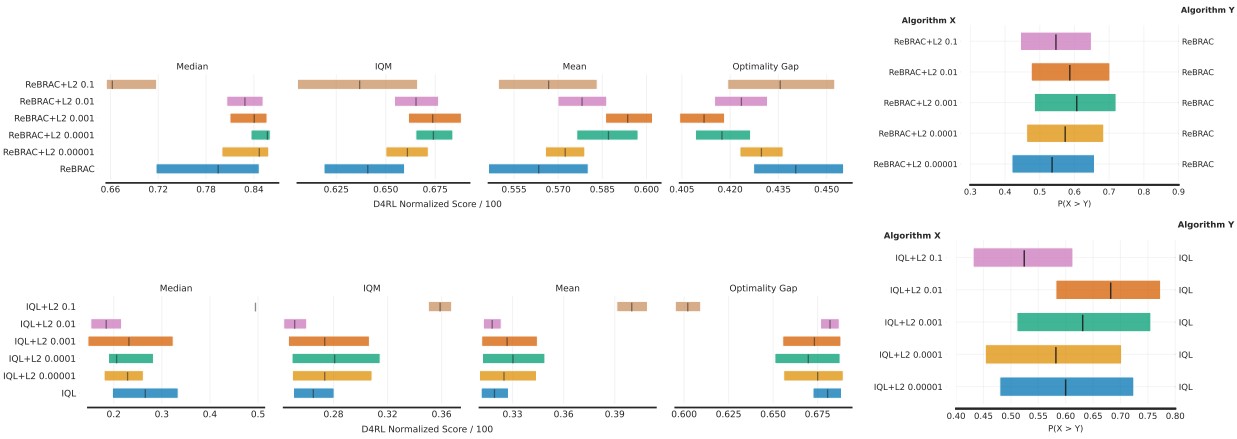

Figure 7: rliable metrics for RAR scores averaged over D4RL subset for different values of $\omega$ parameter with L2 regularization.

## E.3 L1

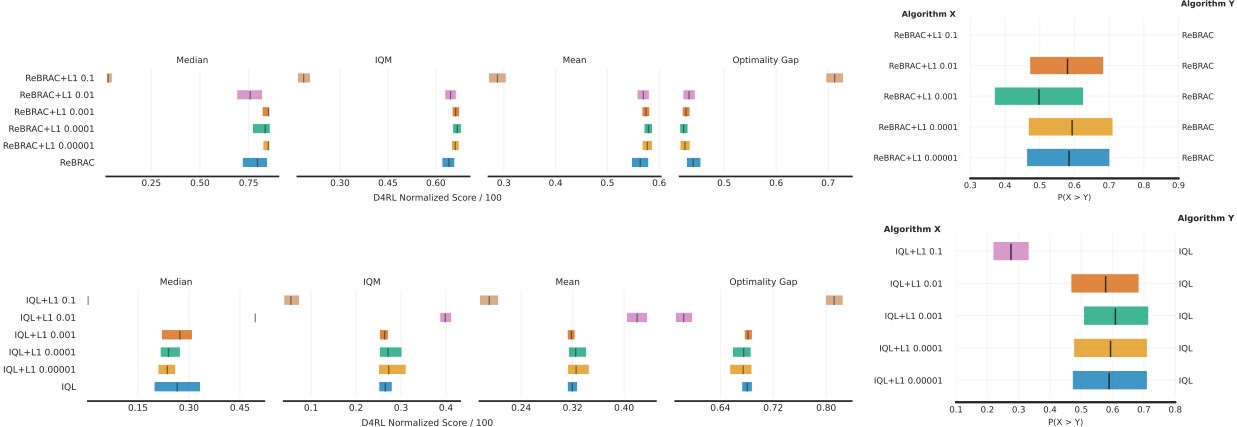

Figure 8: rliable metrics for RAR scores averaged over D4RL subset for different values of $\omega$ parameter with L1 regularization.

## E.4 EN

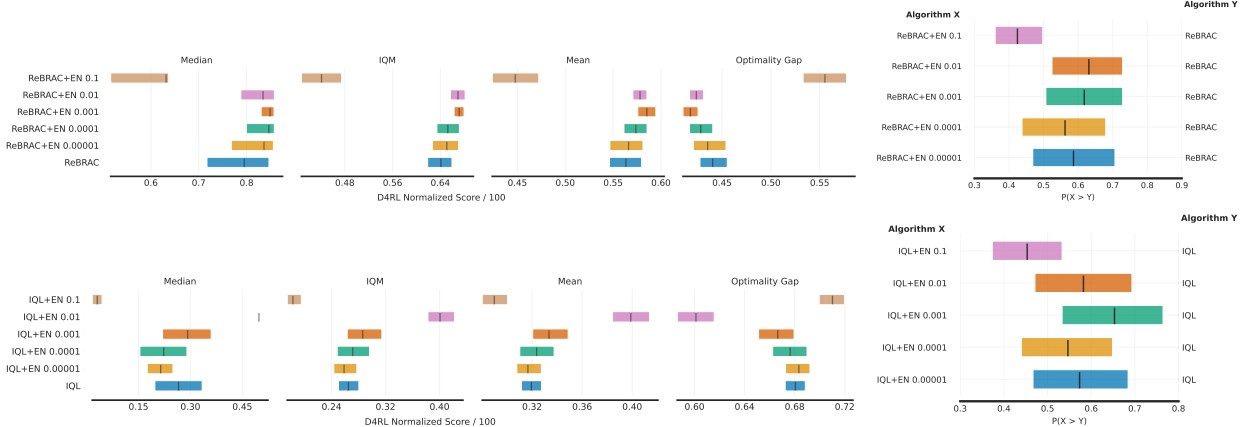

Figure 9: rliable metrics for RAR scores averaged over D4RL subset for different values of $\omega$ parameter with EN regularization.

## E.5 Dropout

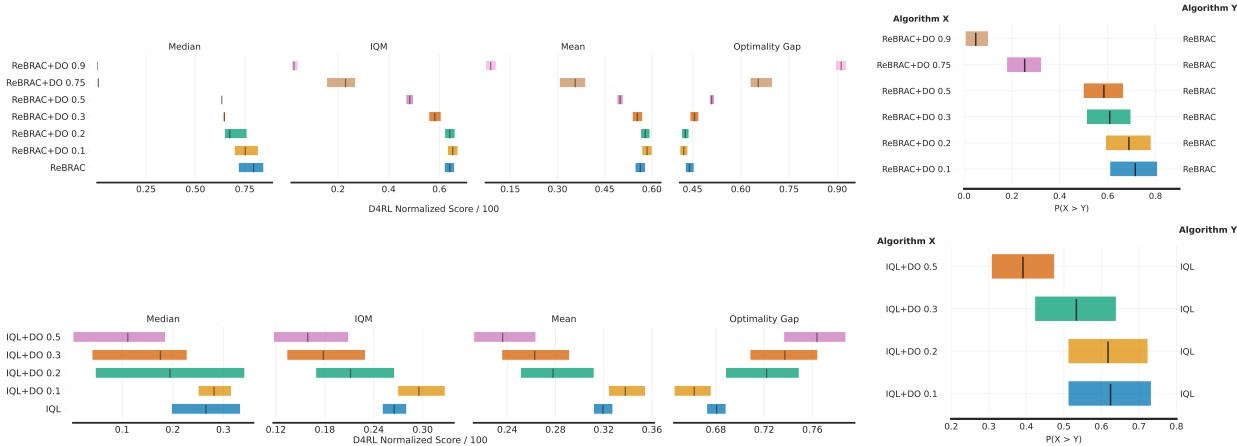

Figure 10: rliable metrics for RAR scores averaged over D4RL subset for different values of dropout rate.

## E.6 Input Noise

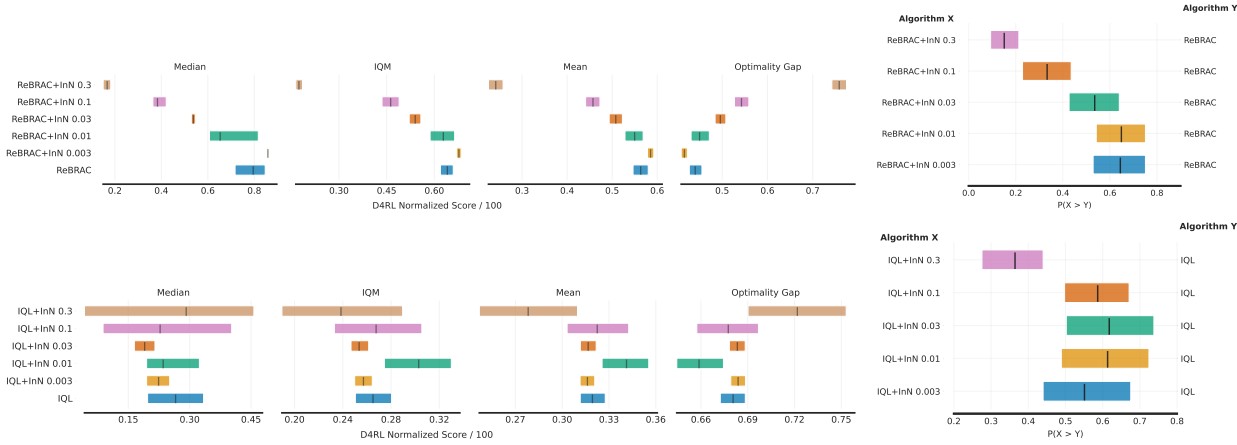

Figure 11: rliable metrics for RAR scores averaged over D4RL subset for different values of $\nu_{InN}$ parameter.

## E.7 Objective Noise

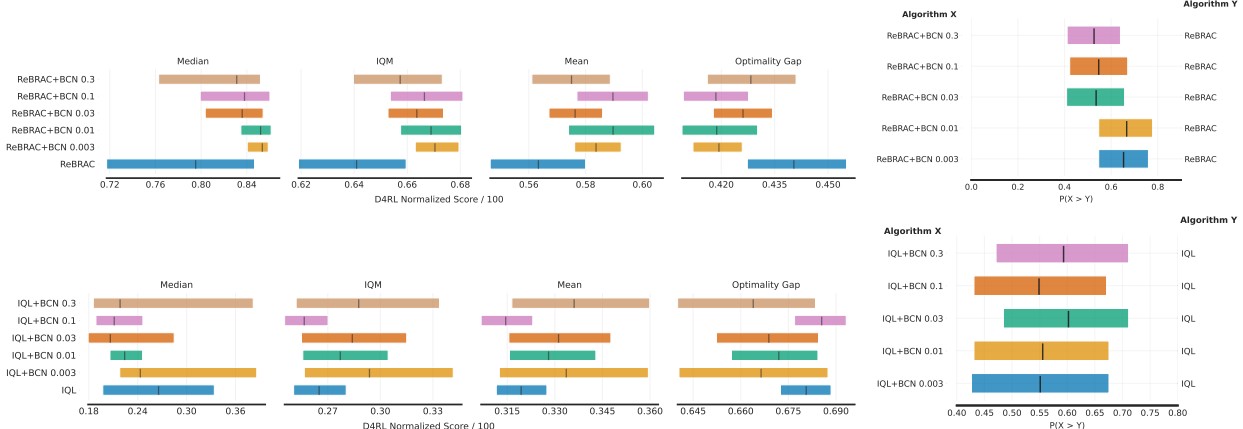

Figure 12: rliable metrics for RAR scores averaged over D4RL subset for different values of $\nu_{BCN}$ parameter.

## E.8 Gradient Noise

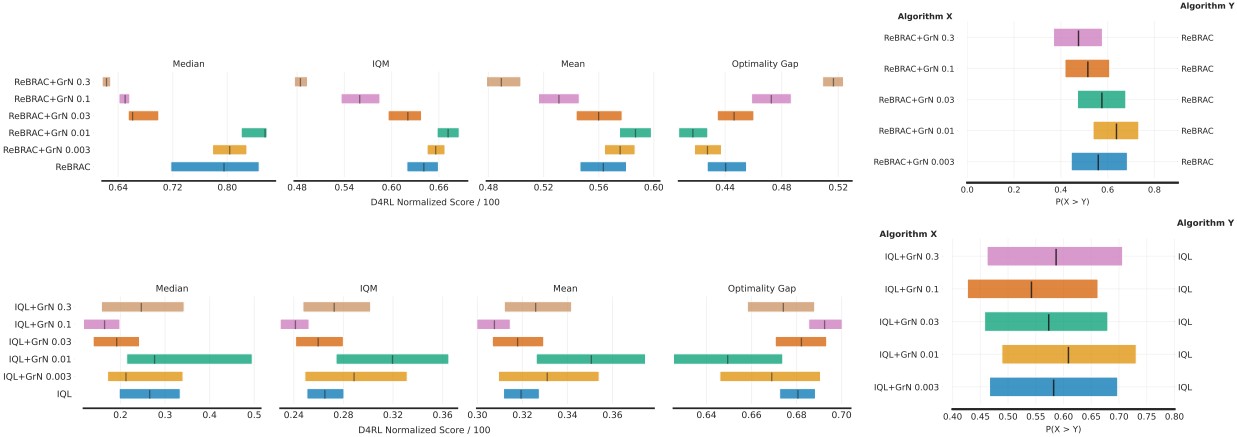

Figure 13: rliable metrics for RAR scores averaged over D4RL subset for different values of $\nu_{GrN}$ parameter.

# F   Probabilities of Improvement for combinations of regularizations

We present rliable (Agarwal et al., 2021) Probabilities of Improvement for combined regularizations in Figure 14. In most of the cases regularizations outperform original algorithms. However, we are not able to see any clear pattern.

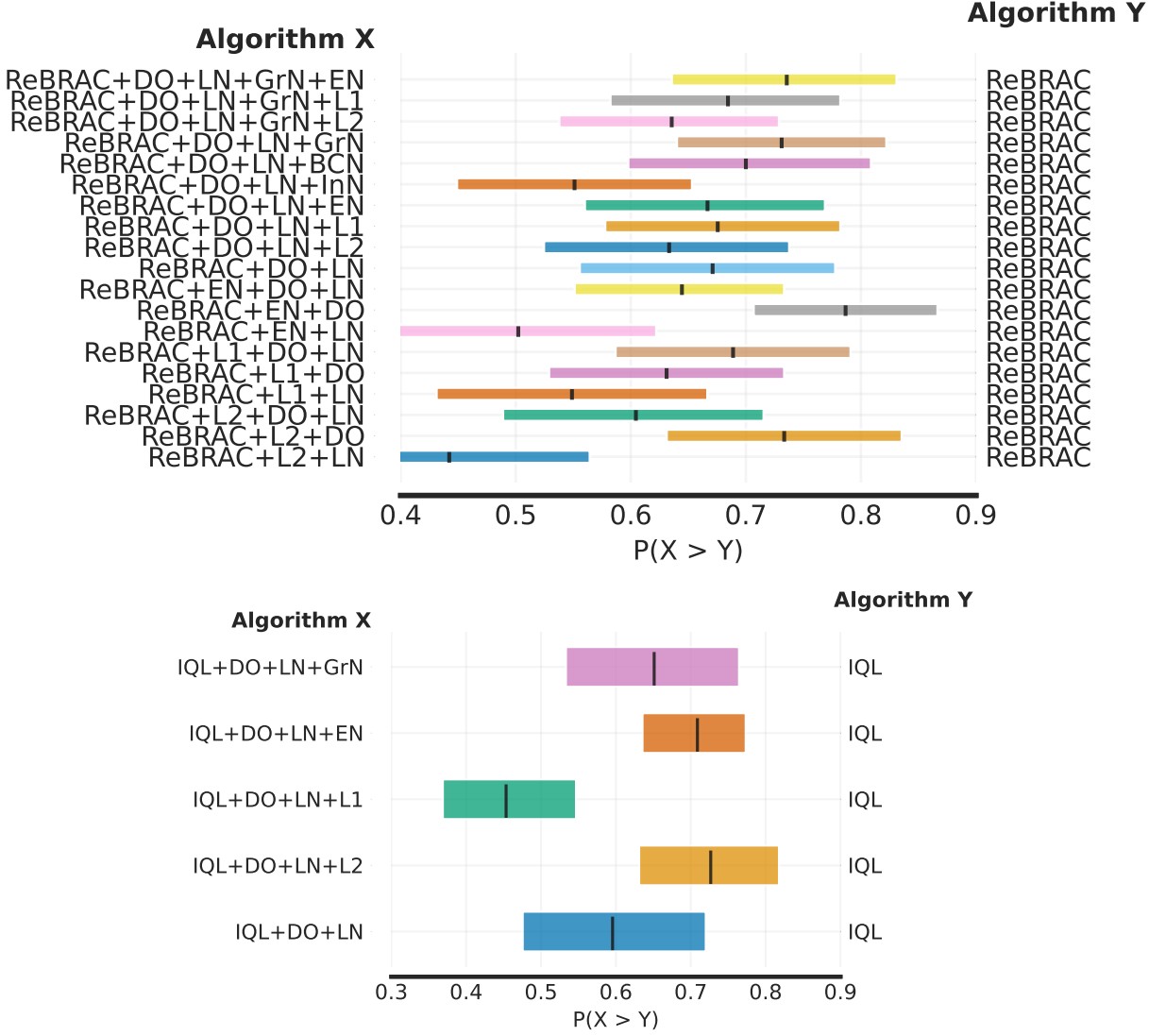

Figure 14: Probabilities of Improvement for RAR scores averaged over D4RL subset when regularizations are combined. Top row ReBRAC performance, bottom row IQL performance. Each task was run over 5 random seeds.

## G   Detailed performance comparison

In this section we provide per-dataset scores for our experiments and compare them against the best solutions on each domain that we could find in literature. We compare against the best ensemble-free (**EF**) approaches and the best ensemble-based (**EB**). Note, in this section we refer to ReBRAC as to the original version of this approach.

Table 6: Final checkpoint mean for Gym-MuJoCo tasks. SAC-DRND(Yang et al., 2024) and RORL(Yang et al., 2022) scores are taken from original works.

| | ReBRAC+CE | | | IQL | | | EF | EB |
|---|---|---|---|---|---|---|---|---|
| Domain | Original | L2 | DO+LN+GrN | Original | L1 | DO+LN+EN | SAC-DRND | RORL |
| hopper-random-v2 | 9.7 ± 1.4 | 8.8 ± 2.0 | 10.4 ± 1.1 | 8.6 ± 5.1 | 8.6 ± 1.5 | 8.7 ± 1.3 | 32.7 ± 0.4 | 31.4 ± 0.1 |
| hopper-medium-v2 | 101.8 ± 1.0 | 102.0 ± 0.2 | 102.3 ± 0.3 | 55.5 ± 4.3 | 53.2 ± 8.2 | 54.9 ± 2.2 | 98.5 ± 1.1 | 104.8 ± 0.1 |
| hopper-expert-v2 | 110.7 ± 0.4 | 110.8 ± 0.6 | 94.9 ± 26.3 | 110.2 ± 1.7 | 108.0 ± 3.0 | 110.2 ± 1.7 | 109.7 ± 0.3 | 112.8 ± 0.2 |
| hopper-medium-expert-v2 | 108.6 ± 6.5 | 107.1 ± 7.2 | 107.3 ± 5.9 | 88.2 ± 17.0 | 91.8 ± 24.9 | 88.2 ± 17.0 | 108.7 ± 0.5 | 112.7 ± 0.2 |
| hopper-medium-replay-v2 | 100.5 ± 4.1 | 91.7 ± 12.8 | 101.8 ± 2.7 | 95.2 ± 7.8 | 92.2 ± 16.2 | 100.9 ± 0.5 | 100.5 ± 1.0 | 102.8 ± 0.5 |
| hopper-full-replay-v2 | 108.4 ± 0.4 | 108.4 ± 0.6 | 108.4 ± 0.5 | 107.3 ± 0.4 | 107.5 ± 0.3 | 107.4 ± 0.1 | 108.2 ± 0.7 | - |
| halfcheetah-random-v2 | 9.4 ± 0.4 | 10.0 ± 0.6 | 10.6 ± 0.5 | 19.1 ± 1.3 | 19.2 ± 0.7 | 18.5 ± 0.9 | 30.4 ± 4.0 | 28.5 ± 0.8 |
| halfcheetah-medium-v2 | 61.6 ± 2.1 | 65.9 ± 3.8 | 64.8 ± 1.8 | 49.8 ± 0.2 | 49.8 ± 0.3 | 49.8 ± 0.7 | 68.3 ± 0.2 | 66.8 ± 0.7 |
| halfcheetah-expert-v2 | 104.4 ± 2.0 | 103.7 ± 1.7 | 102.4 ± 5.1 | 96.9 ± 0.2 | 96.6 ± 0.3 | 96.3 ± 1.4 | 106.2 ± 3.7 | 105.2 ± 0.7 |
| halfcheetah-medium-expert-v2 | 103.3 ± 2.4 | 103.6 ± 2.1 | 105.2 ± 1.6 | 93.8 ± 1.0 | 91.4 ± 4.1 | 92.4 ± 2.5 | 108.5 ± 1.1 | 107.8 ± 1.1 |
| halfcheetah-medium-replay-v2 | 54.3 ± 2.3 | 54.0 ± 2.2 | 56.9 ± 2.7 | 45.0 ± 0.6 | 45.3 ± 0.4 | 45.8 ± 0.3 | 52.1 ± 4.8 | 61.9 ± 1.5 |
| halfcheetah-full-replay-v2 | 80.9 ± 3.2 | 82.2 ± 2.7 | 82.1 ± 2.5 | 75.6 ± 0.5 | 75.9 ± 0.4 | 75.6 ± 0.3 | 81.4 ± 1.7 | - |
| walker2d-random-v2 | 10.9 ± 7.0 | 7.2 ± 5.1 | 9.8 ± 6.0 | 6.9 ± 4.2 | 6.5 ± 6.1 | 6.9 ± 4.2 | 21.7 ± 0.1 | 21.4 ± 0.2 |
| walker2d-medium-v2 | 84.4 ± 4.5 | 85.5 ± 1.2 | 86.1 ± 0.7 | 82.7 ± 2.2 | 83.4 ± 0.6 | 83.5 ± 0.3 | 95.2 ± 0.7 | 102.4 ± 1.4 |
| walker2d-expert-v2 | 112.5 ± 0.3 | 114.5 ± 0.9 | 113.9 ± 0.5 | 113.8 ± 0.3 | 114.5 ± 0.6 | 114.8 ± 0.5 | 114.0 ± 0.5 | 115.4 ± 0.5 |
| walker2d-medium-expert-v2 | 110.2 ± 0.2 | 110.5 ± 0.8 | 111.6 ± 0.5 | 112.8 ± 0.6 | 112.5 ± 1.1 | 113.0 ± 1.0 | 109.6 ± 1.0 | 121.2 ± 1.5 |
| walker2d-medium-replay-v2 | 88.3 ± 8.3 | 81.3 ± 17.6 | 92.7 ± 3.1 | 84.0 ± 4.8 | 75.9 ± 9.7 | 79.1 ± 6.6 | 91.0 ± 2.9 | 90.4 ± 0.5 |
| walker2d-full-replay-v2 | 107.1 ± 6.1 | 107.5 ± 2.7 | 110.0 ± 5.1 | 97.6 ± 1.8 | 98.5 ± 1.2 | 99.8 ± 2.5 | 109.6 ± 0.7 | - |
| Gym-MuJoCo average | 81.5 | 80.8 | 81.7 | 74.6 | 73.8 | 74.8 | 85.9 | 85.7 |

Table 7: Final checkpoint mean for AntMaze tasks. TD3-BC-N (Beeson & Montana, 2024) scores are taken from the original work. ReBRAC+CE also stands for **EF** here.

| | ReBRAC+CE | | | IQL | | | EB |
|---|---|---|---|---|---|---|---|
| Domain | Original | L2 | DO+LN+GrN | Original | L1 | DO+LN+EN | TD3-BC-N |
| antmaze-umaze-v2 | 99.1 ± 0.9 | 99.4 ± 0.5 | 99.2 ± 0.7 | 78.6 ± 6.3 | 77.4 ± 6.5 | 73.8 ± 6.6 | 98.3 ± 0.7 |
| antmaze-umaze-diverse-v2 | 93.2 ± 3.9 | 92.2 ± 5.8 | 95.9 ± 2.0 | 68.7 ± 4.6 | 70.7 ± 6.9 | 74.7 ± 3.4 | 90.6 ± 1.3 |
| antmaze-medium-play-v2 | 90.5 ± 4.3 | 86.7 ± 8.2 | 93.9 ± 4.0 | 72.7 ± 2.3 | 75.7 ± 5.2 | 75.1 ± 6.7 | 87.0 ± 1.5 |
| antmaze-medium-diverse-v2 | 91.8 ± 3.1 | 90.9 ± 5.0 | 92.2 ± 5.5 | 74.9 ± 4.8 | 72.2 ± 6.9 | 72.6 ± 6.0 | 86.2 ± 1.5 |
| antmaze-large-play-v2 | 85.7 ± 7.1 | 86.1 ± 4.8 | 88.3 ± 4.2 | 43.2 ± 8.1 | 43.0 ± 7.0 | 47.7 ± 6.8 | 76.2 ± 1.9 |
| antmaze-large-diverse-v2 | 81.6 ± 10.5 | 87.0 ± 6.0 | 86.3 ± 4.0 | 28.3 ± 8.9 | 33.3 ± 8.5 | 52.1 ± 5.2 | 74.2 ± 2.0 |
| AntMaze average | 90.3 | 90.4 | 92.6 | 61.1 | 62.1 | 66.0 | 85.5 |

Table 8: Final checkpoint mean for Adroit tasks. ReBRAC+CE* stands for ReBRAC+CE without hyperparameters consistency. ReBRAC+CE*(Tarasov et al., 2024a) and RORL(Yang et al., 2022) scores are taken from original works.

| Domain | ReBRAC+CE | | | IQL | | | EF | EB |
| | Original | L2 | DO+LN+GrN | Original | L1 | DO+LN+EN | ReBRAC+CE* | RORL |
|---|---|---|---|---|---|---|---|---|
| pen-human-v1 | $51.6 \pm 10.6$ | $71.7 \pm 12.2$ | $91.2 \pm 20.4$ | $20.4 \pm 26.3$ | $107.4 \pm 9.4$ | $61.8 \pm 19.2$ | $93.8 \pm 15.4$ | $33.7 \pm 7.6$ |
| pen-cloned-v1 | $97.6 \pm 13.7$ | $94.3 \pm 18.6$ | $88.6 \pm 10.7$ | $5.3 \pm 13.0$ | $90.4 \pm 15.1$ | $0.6 \pm 4.6$ | $106.1 \pm 28.6$ | $35.7 \pm 35.7$ |
| pen-expert-v1 | $153.6 \pm 4.0$ | $150.5 \pm 6.4$ | $153.1 \pm 6.4$ | $73.0 \pm 38.5$ | $147.3 \pm 4.1$ | $148.1 \pm 2.2$ | $158.2 \pm 0.8$ | $130.3 \pm 4.2$ |
| door-human-v1 | $-0.1 \pm 0.1$ | $-0.1 \pm 0.1$ | $0.0 \pm 0.1$ | $5.1 \pm 2.1$ | $11.4 \pm 1.5$ | $7.5 \pm 4.6$ | $0.0 \pm 0.1$ | $3.7 \pm 0.7$ |
| door-cloned-v1 | $0.3 \pm 0.3$ | $0.3 \pm 0.3$ | $5.0 \pm 7.4$ | $0.1 \pm 0.0$ | $0.6 \pm 1.6$ | $4.5 \pm 1.4$ | $0.0 \pm 0.0$ | $-0.1 \pm 0.1$ |
| door-expert-v1 | $106.2 \pm 0.2$ | $105.6 \pm 1.4$ | $106.2 \pm 0.3$ | $105.9 \pm 1.9$ | $103.2 \pm 2.7$ | $104.3 \pm 3.2$ | $106.2 \pm 0.2$ | $104.9 \pm 0.9$ |
| hammer-human-v1 | $0.6 \pm 0.4$ | $2.8 \pm 3.7$ | $0.8 \pm 0.7$ | $2.3 \pm 1.4$ | $2.6 \pm 1.5$ | $2.5 \pm 1.7$ | $2.8 \pm 4.1$ | $2.3 \pm 2.3$ |
| hammer-cloned-v1 | $3.7 \pm 4.3$ | $6.4 \pm 9.5$ | $4.7 \pm 7.8$ | $0.3 \pm 0.0$ | $0.5 \pm 0.3$ | $1.1 \pm 1.0$ | $4.0 \pm 3.8$ | $1.7 \pm 1.7$ |
| hammer-expert-v1 | $125.6 \pm 19.3$ | $132.0 \pm 5.7$ | $135.0 \pm 0.3$ | $129.6 \pm 0.4$ | $128.6 \pm 3.0$ | $129.6 \pm 0.4$ | $134.3 \pm 0.3$ | $132.2 \pm 0.7$ |
| relocate-human-v1 | $-0.1 \pm 0.0$ | $0.2 \pm 0.4$ | $0.2 \pm 0.4$ | $0.1 \pm 0.0$ | $0.2 \pm 0.3$ | $0.2 \pm 0.3$ | $0.1 \pm 0.3$ | $0.0 \pm 0.0$ |
| relocate-cloned-v1 | $0.3 \pm 0.4$ | $0.2 \pm 0.5$ | $0.3 \pm 0.3$ | $0.1 \pm 0.1$ | $0.1 \pm 0.2$ | $0.1 \pm 0.1$ | $0.3 \pm 0.3$ | $0.0 \pm 0.0$ |
| relocate-expert-v1 | $98.7 \pm 10.0$ | $102.5 \pm 5.4$ | $101.6 \pm 7.0$ | $108.0 \pm 0.9$ | $108.8 \pm 0.6$ | $107.7 \pm 1.7$ | $106.7 \pm 4.3$ | $3.4 \pm 4.5$ |
| Adroit w\o expert average | 19.2 | 22.1 | 23.8 | 4.2 | 26.7 | 9.8 | 25.8 | 9.6 |
| Adroit average | 53.2 | 55.6 | 57.2 | 37.5 | 58.4 | 47.3 | 59.3 | 41.0 |

# H  Per-domain actor generalization

Here we provide experimental results from subsection 4.4 individually per D4RL domain. The only surprising observation comes from Figure 17 where the addition of regularizations reduces IQL robustness to noise perturbations.

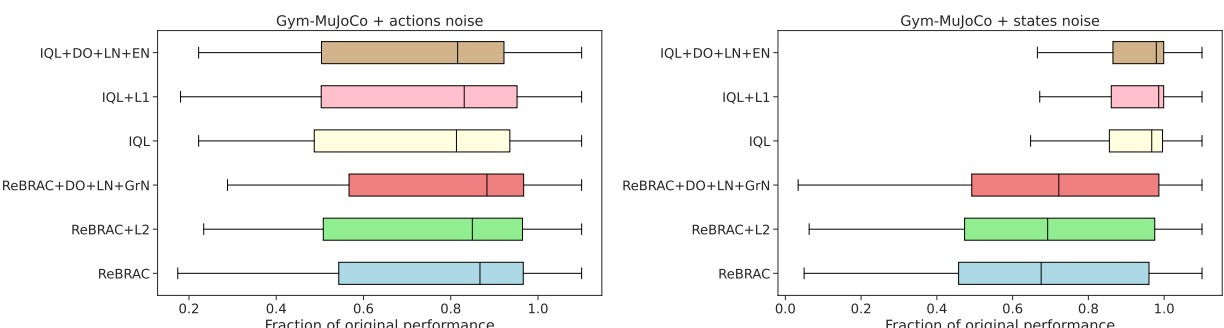

Figure 15: Final checkpoint performance with noise injection divided by the noise-free performance on Gym-MuJoCo tasks.

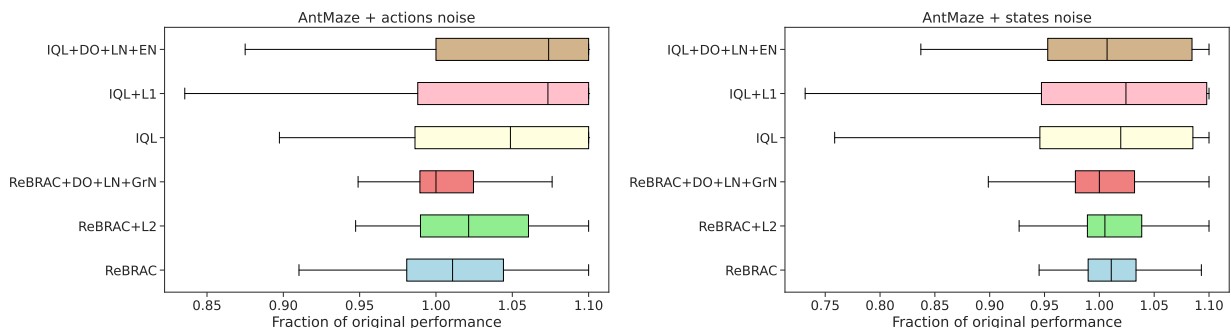

Figure 16: Final checkpoint performance with noise injection divided by the noise-free performance on AntMaze tasks.

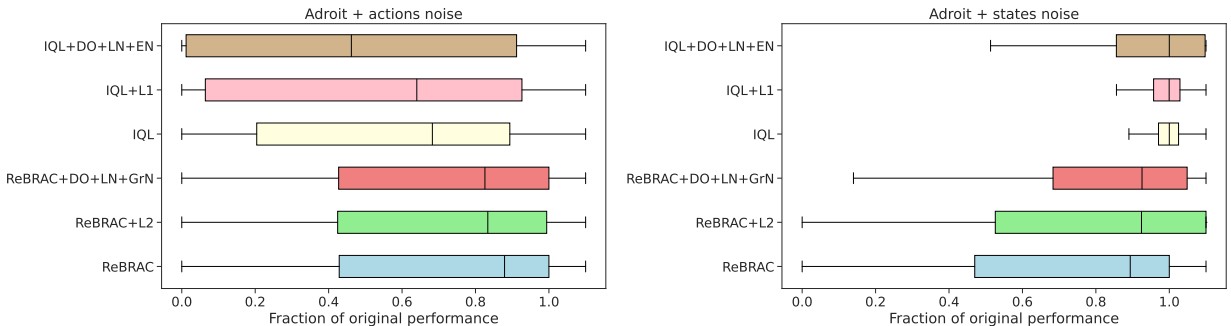

Figure 17: Final checkpoint performance with noise injection divided by the noise-free performance on Adroit tasks.

# I Per-domain actor internal metrics

In this section we provide results from subsection 4.5 per D4RL domains.

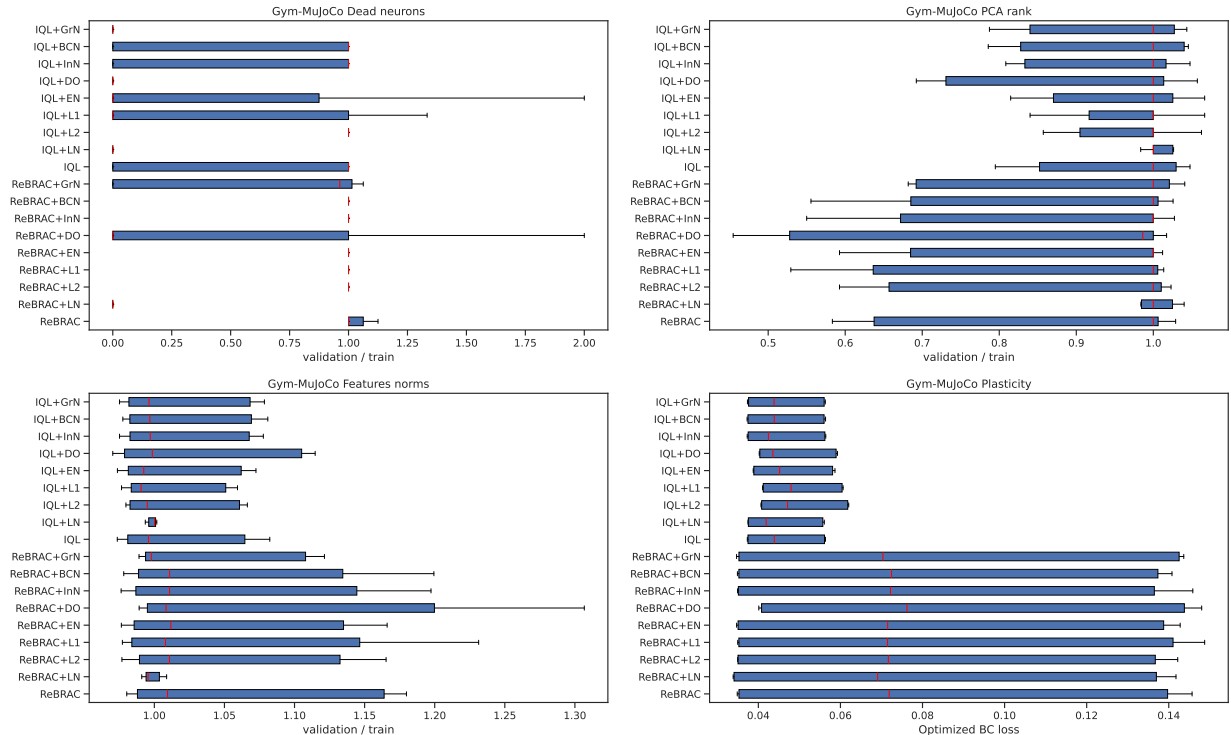

Figure 18: Plasticity and validation data metrics divided by the train data metrics from actor penultimate linear layer of the final checkpoint averaged over subset of Gym-MuJoCo tasks.

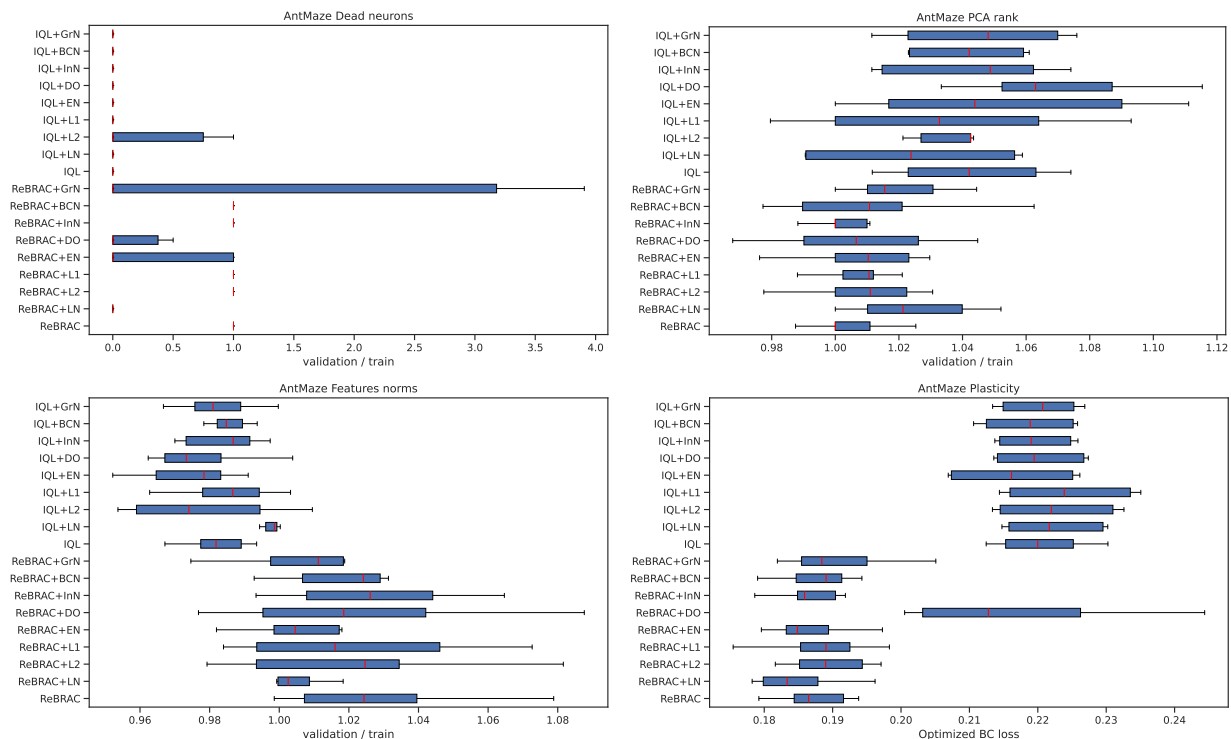

Figure 19: Plasticity and validation data metrics divided by the train data metrics from actor penultimate linear layer of the final checkpoint averaged over subset of AntMaze tasks.

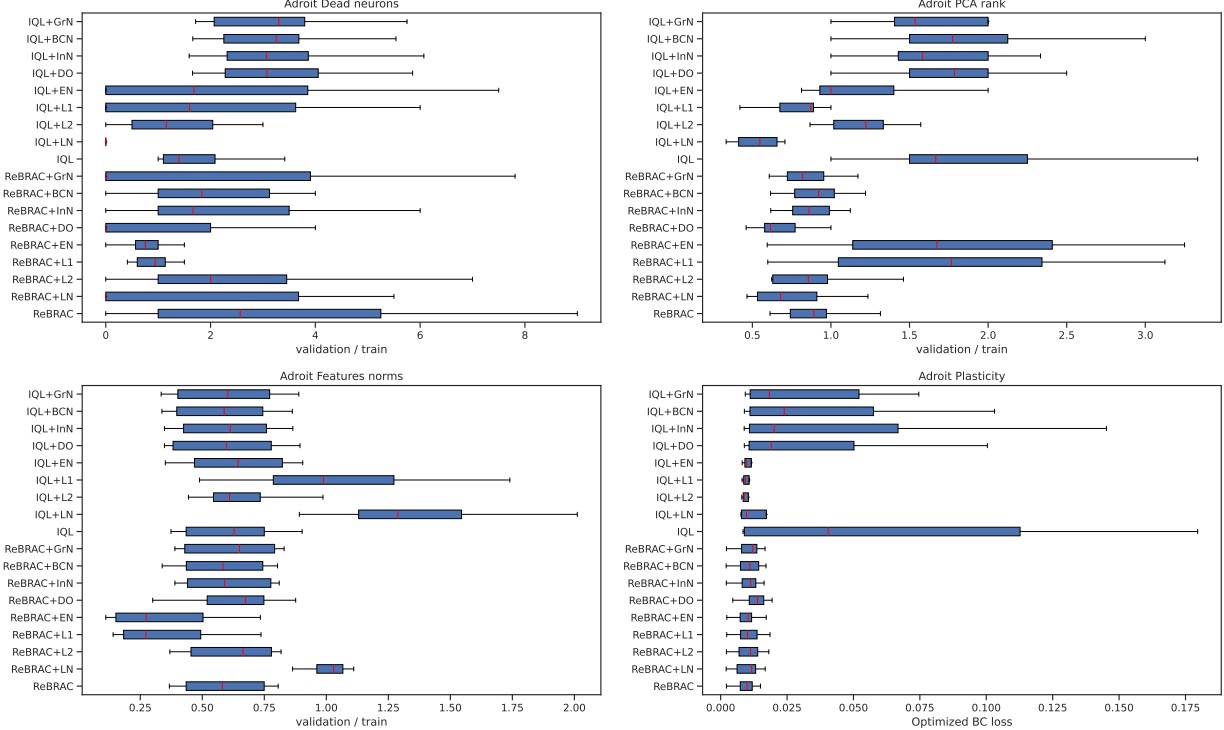

Figure 20: Plasticity and validation data metrics divided by the train data metrics from actor penultimate linear layer of the final checkpoint averaged over subset of Adroit tasks.

