# OpenReview forum: "The Role of Deep Learning Regularizations on Actors in Offline RL"
_TMLR — Rejected by TMLR_

### Review · Reviewer_Nmm2 · 2024-10-08

**Summary Of Contributions:**

This paper proposed to empirically study various network regularization techniques in the actor network of reinforcement learning actor-critic methods. The topic is interesting and has not been widely studied before. It could be of broad interest to the community. The authors conducted systematic comparison between different regularization techniques and their combinations.

**Audience:**

No

**Broader Impact Concerns:**

ethical concerns not applicable

**Claims And Evidence:**

No

**Requested Changes:**

please see the weakness section for questions and requested changes.

**Strengths And Weaknesses:**

**Strengths:**\
This paper investigated an interesting but less popular topic which could be of broad interest to the RL community. The analogy between supervised learning where network regularization tricks show great promise and the offline RL is interesting. Empiricallly, the paper systematically compared many techniques and showed consistent improvement over the original algorithms.

**Weaknesses**:\
However, I cannot recommend publication of this version as I believe many important parts are missing. Below lists my points:
1. I think the authors need to clearly position this paper since it is an RL paper. This is because in RL many algorithms are based on the concept of *entropy regularization* that softens the actor, see refs [1-4]. This paper also focused on techniques applied to the actor network (I would call them hard regularization). Therefore, it seems necessary to clarify and discuss this difference between the soft and hard regularization. This discussion could be helpful in e.g. section 4.4 about generalization: as the authors pointed out, no significant generalization improvement was observed. However, by soft regularization, we can expect improved generalization in terms of noise tolerance, see ref [2]. What could lead to this difference?

2. The number of algorithms seems to be very limited. More importantly, it is not clear to me why these two algorithms were picked, what consideration they reflected.  I would expect a pool that can reflect the diversity of soft regularization:  TD3BC, AWAC, TAWAC, InAC, IQL,  see refs [5-9]. It would be much more convincing and interesting to see how the hard regularization techniques would behave on top of these soft regularized algorithms.

3. Section 4.4 and 4.5 lack in-depth discussion. While I appreciate the honest report that the hard regularization did not lead to significant improvement, I believe more details are needed. For example, was the similar performance due to similar action sequences or due to different sequences but similar final accumulated rewards? If it was the first case, I wonder what regularization could lead to a very generalizable actor? Because the actor network outputs probability density, how tricks like dropout could significantly change its property like shape, mode, etc?  And does this linked to the distribution used, ie., not just the Gaussian, but also other choices (see refs [10,12])?

4. As an empirical study, 5 seeds are too limited. And we can indeed observe large variance in the plots. I would suggest to at least run 10 seeds for statistical significance.

References:\
[1] soft actor-critic\
[2] leverage the average an analysis of kl regularization in reinforcement learning\
[3] Investigating Forward and Reverse KL Divergences\
[4] Generalized Munchausen Reinforcement Learning using Tsallis KL Divergence\
[5] A Minimalist Approach to Offline Reinforcement Learning\
[6] AWAC: Accelerating Online Reinforcement Learning with Offline Datasets\
[7] Offline Reinforcement Learning via Tsallis Regularization\
[8] The In-Sample Softmax for Offline Reinforcement Learning\
[9] Offline Reinforcement Learning with Implicit Q-Learning\
[10] Improving Stochastic Policy Gradients in Continuous Control with Deep Reinforcement Learning using the Beta Distribution\
[11] quasi-optimal reinforcement learning with continuous actions\
[12] q exponential family policy optimization

---

> ### Author Response · Authors · 2024-10-19
>
> We would like to thank the reviewer for their thoughtful and constructive feedback. Below, we address each of the points raised in detail:
>
> # Addressing weaknesses
> 1. We appreciate the reviewer's suggestion regarding the distinction between soft and hard regularizations. However, we consider this distinction somewhat out of the scope of our current study. Soft regularizations, such as entropy regularization, are typically employed in online RL to facilitate exploration or to stabilize policy updates by preventing abrupt changes due to non-stationary input distributions. In offline RL, these challenges are less prominent, as we are working with static datasets, thus mitigating the need for exploration and stabilizing updates. Given that our focus is on the application of deep learning regularization techniques (i.e., "hard" regularization) to actor networks, we did not delve into the comparative analysis with soft regularizations.
> 2. As noted in subsection 3.1, our selection was guided by recent work by Park et al. (https://arxiv.org/abs/2406.09329) and Tarasov et al. (https://arxiv.org/abs/2210.07105), where ReBRAC and IQL were identified as highly effective in the offline RL setting. Our work is heavily inspired by Park et al., where they conduct the entire study using ReBRAC-like algorithm and IQL as this algorithms were shown to be the most effective for offline RL. That claim was earlier supported by Tarasov et al. where authors benchmarked the wide range of algorithms across many different tasks. We actually used 2 algorithms from the pool you provided: IQL and ReBRAC which is a modification of TD3+BC (we will make this fact more explicit in the updated script).  While AWAC is similar to IQL, the latter consistently outperformed it in existing studies, which is why we did not include AWAC in our experiments.
> 3. We understand the request for more in-depth analysis of Sections 4.4 and 4.5 but we are unable to run any more experiments. Due to the continuous action space in our benchmark, it is challenging to analyze specific action sequences directly. Our main hypothesis was inspired by Park et al. (https://arxiv.org/abs/2406.09329), who identified a lack of actor generalization as a key limitation in offline RL, particularly with OOD inputs. We tested whether deep learning regularizations, which have proven beneficial in non-RL contexts, could alleviate this issue. While we observed performance improvements, our findings suggest that these improvements are likely due to smoother optimization or some other aspects rather than direct enhancements in generalization.
> 4. We acknowledge the reviewer's concern regarding the limited number of seeds (five) used in our experiments. Unfortunately, due to the scale of our study and the computational resources available, we were unable to run additional seeds. Nonetheless, we recognize that this is a common challenge in the offline RL community. For instance, popular works such as CQL (https://arxiv.org/abs/2006.04779), TD3+BC (https://arxiv.org/abs/2106.06860), SAC-N (https://arxiv.org/abs/2110.01548), and DT (https://arxiv.org/abs/2106.01345) report experiments with as few as three to five seeds on the same benchmark. To address this limitation, we adopted rliable metrics (https://arxiv.org/abs/2108.13264), which are designed to ensure statistical significance even with a small number of seeds. These metrics, combined with confidence intervals in our plots, demonstrate clear and statistically significant benefits of regularization in cases where the intervals do not overlap (e.g., weight decay in IQL, Figure 1).
>
> We hope that our responses sufficiently address the reviewer's concerns and we look forward to any further suggestions. Thank you again for your feedback.

---

> > ### Comment · Reviewer_Nmm2 · 2024-10-21
> > **more discussion needed**
> >
> > Hi, thanks for the reply.
> >
> > I think it is okay if you cannot run more experiments. But I do believe that more discussion is needed. Currently the paper has only 10 pages and I think it is acceptable to fill in until 12 pages which still falls into short paper of TMLR.
> >
> > Specifically, I think the following points need to be discussed more:
> > 1. could you provide more details from the existing experimental results?
> > >  For example, was the similar performance due to similar action sequences or due to different sequences but similar final accumulated rewards?
> >
> > 2. could you elaborate more on what regularization could lead to a very generalizable actor? For example, is it reasonable to conjecture that dropout forced the actor network to be robust against small perturbances?
> >
> > 3. let me clarify more on entropy regularization. They include not just entropy but also divergences and some technical results apply to the offline context as well. One of the most popular directions is to regularize the actor by some divergence so the learned policy stay close to the behavior policy, and often has a closed-form expression e.g. the Boltzmann in the case of KL regularization. These are soft regularizations because they soften the actor. Your investigation was on the network directly which I would call hard regularization. They both care about the actor, and improving robustness, generalization, etc, and they actually can be run together without any issue.  It is okay to accept the current result as it is, but I think this difference deserves a paragraph to emphasize this work is complementary to the existing soft regularization work.
> >
> > 4. I cannot find what was the policy parametrization in the paper, I would assume it was Gaussian, i.e. the actor network output $\mu$ and $\sigma$ that parametrized a Gaussian policy. I think it would be helpful to add a paragraph to related work, just to be more considerate of recent work on policy parametrizations, since it is still unclear whether the tricks investigated in the paper would be different for other policy functionals, such as Beta, $q$-Gaussian, etc.

---

> ### Author Response · Authors · 2024-10-27
>
> 1. Unfortunately, as we mentioned before, we are unable to provide more insights from the experiments, e.g. analyzing sequences does not provide any interpretable results due to the continuous nature of the environments.
> 2. It is an open question and in our study we try to understand whether common regularizations from deep learning field allow us to build more generalizable policies. Our results show that most probably it does not have the desired effect in terms of generalization, however provides benefits in some other aspects.
> 3. We provide results for ReBRAC which includes the soft regularization (forcing the policy to stay close to dataset with BC term) if we understand your explanation correctly. We will consider adding the paragraph about soft regularizations.
> 4. For IQL parametrization is Gaussian. ReBRAC only predicts the action as it is done in DDPG/TD3.

---

### Review · Reviewer_g1nd · 2024-10-17

**Summary Of Contributions:**

The goal of this paper is to investigate the effects of regularisation techniques on the actor in offline-RL algorithms. The motivation of this being that regularisation has only been studied and generally applied to the learning of the critic/value functions, but its affect on the actor has not been explored or understood. The main contributions to this work are:
- Answering if regularisation on the actor is useful in the offline RL setting
- How Is it useful? And how does regularisation impact/affect the actor?

Other algorithmic contributions studied are:
- Hyperparameter sensitivity of the algorithms when applying regularization
- How different regularisations can be combined and applied to existing offline RL algorithms.

**Audience:**

Yes

**Broader Impact Concerns:**

No major concerns.

**Claims And Evidence:**

Yes

**Requested Changes:**

No major changes. I had some minor suggestions for improvement for the authors in the previous section.

**Strengths And Weaknesses:**

Strengths:
This is a very well written paper. The presentation and structure of the contributions, motivation and the obtained results are very clear and to the point. Background and prior work is covered well and is relatively well covered. Choice of baseline algorithms are clearly explained and justified based on prior literature. Results are also presented very clearly and discussed well. The paper answers the questions it set out to study and covers a gap in understanding of Deep RL literature which is the application of regularisation methods on the actor networks. Very much appreciate the very clear and factual reporting style in the results as well and pointing out the limitations of applying regularisation (i.e. does not help policy generalisation, hyperparameter sensitivity)

Weaknesses:
Abit nitpicky but would make things slightly clearer:
- some questions posed that this paper answers in this work is abit repetitive. I think the contributions are valid and fair but would benefit from being clearer and straight to the point. Point 1 and point 3 seem to be very related where point 1 is the more general version that covers point 3.
- to help keep the paper abit more self-contained and complete. a nice addition would be to add a short summary/explanation (psuedocode) of the two algorithms studied ReBRAC and IQL either in the appendix with a reference in the main text in section 3.1. and clearly pointing out the actor networks in those algorithms where the regularization is applied, and also the presence of regularization in other parts of the algorithms.

---

> ### Author Response · Authors · 2024-10-19
>
> We thank reviewer a lot for their feedback.
>
> # Addressing weaknesses
> 1. We agree that points 1 and 3  are related. However, from our point of view, it is important to highlight point 3 separately as validation of the generalization improvement is one of the central parts of our study.
> 2. Thank you for the idea. We are adding short descriptions of the algorithms and pseudocodes for regularizations usages in our updated script in Appendix. We will notify reviewers about the update once it is done.

---

### Review · Reviewer_ZAs5 · 2024-10-18

**Summary Of Contributions:**

The paper presents a rigorous empirical study on the actor network architecture in offline RL. The results reported focus primarily on ReBRAC, which is considered a state-of-the-art offline RL method. While the effects of regularization techniques such as LayerNorm and Dropout have been implicitly suggested in previous studies, I am not aware of any prior work that has thoroughly investigated and summarized their impact. Clarifying the effects of these techniques on improving the performance of offline RL methods would be beneficial for both researchers and practitioners.

**Audience:**

Yes

**Broader Impact Concerns:**

not applicable.

**Claims And Evidence:**

Yes

**Requested Changes:**

- For completeness, I recommend including a brief description of IQL and ReBRAC, such as their objective functions, in the preliminary section.

- I do not clearly understand how normalization was applied in the experiments. While several normalization techniques were introduced in the first section of page 3, it is unclear which normalization technique was used and how they were combined. I do not see any results regarding the effects of the individual normalization techniques. Please add them if the authors have conducted the experiments regarding the effects of the individual normalization techniques, such as Layer Normalization , Feature Normalization, Group Normalization, and Spectral Normalization.

- Please consider adding a small set of experiments with TD3+BC, as this would help clarify the relationship between conservative critic learning and regularization techniques.

- Regarding plasticity, I do not clearly understand how it is measured based on the behavior cloning term. Please elaborate on how plasticity was quantified and why this is a suitable method for measuring it. If there is existing work that rigorously justifies how to measure the plasticity of the actor, please provide the reference.

**Strengths And Weaknesses:**

Strengths of the paper:

- The paper reports the results of investigating the effects of several techniques, including Layer Normalization, Spectral Normalization, Dropout, and Weight Decay. It is also interesting that the study not only examines the effect of each individual technique, but also investigates the effects when they are combined.

- The finding that weight decay is effective in IQL and that L2 regularization is effective in ReBRAC is also intriguing.

Weaknesses of the paper:

- While ReBRAC demonstrates strong performance, the results are limited to only ReBRAC and IQL. ReBRAC is built upon TD3+BC, but its critic objective includes a penalty for deviation from the actions in the dataset. Including a smaller set of experiments with TD3+BC would help clarify the relationship between conservative critic learning and regularization techniques.

- There are some aspects of the presentation that could be improved (see the 'Requested Changes' section for details).

---

> ### Author Response · Authors · 2024-10-19
>
> We thank reviewer for their work
>
> # Addressing Requested Changes
> 1. We are including suggested descriptions in the updated version of the script. Thank you for the suggestion.
> 2. We used only one normalization a time when used any. And when combining with other regularizations we try only LayerNorm to keep experiments tractable. Clarification on that will be added in the subsection 3.2 of the updated script.
> 3. Unfortunately, we are unable to conduct any additional experiments. However, ReBRAC's critic regularization does not play a significant role in algorithm's performance as shown by the authors of ReBRAC. So we assume that TD3+BC experiments will not provide further insights.
> 4. Plasticity loss is an inability of the trained network to update it's predictions using the new data. We measure this by optimizing the trained actor with simple BC loss (actions MSE) over 100 updates using the validation dataset different from the training one. Final loss value demonstrates the plasticity of the network: higher loss values indicate the lack of the adaptation ability. This metric is also good as it provides a single scalar value which can be compared among different modifications and it is not dependent on anything beside the action space which is simillar across D4RL domains. We are adding some comments on that in Appendix A.

---

> > ### Comment · Reviewer_ZAs5 · 2024-10-21
> > **Follow-up questions**
> >
> > > We used only one normalization a time when used any. And when combining with other regularizations we try only LayerNorm to keep experiments tractable. Clarification on that will be added in the subsection 3.2 of the updated script.
> >
> > Thank you for the clarification. While Section 3.2 mentions that LayerNorm, FeatureNorm, GroupNorm, and SpectralNorm were considered, the results for FeatureNorm, GroupNorm, and SpectralNorm only appear in Appendix E.1. I suggest moving these results from Appendix E.1 to the main text, for example, by combining them with the results in Figure 1. Additionally, why are the results for IQL with SpectralNorm missing from Appendix E.1?
> >
> > >  However, ReBRAC's critic regularization does not play a significant role in algorithm's performance as shown by the authors of ReBRAC. So we assume that TD3+BC experiments will not provide further insights.
> >
> > Thank you for the response. I would like to ask a question to confirm my understanding. I reviewed the [original ReBRAC paper](https://proceedings.neurips.cc/paper_files/paper/2023/file/26cce1e512793f2072fd27c391e04652-Paper-Conference.pdf).
> > In my understanding, the only algorithmic difference between TD3+BC and ReBRAC is the critic penalty. The third row from the bottom in Table 8 of the  [original ReBRAC paper](https://proceedings.neurips.cc/paper_files/paper/2023/file/26cce1e512793f2072fd27c391e04652-Paper-Conference.pdf) shows that the critic penalty does not significantly affect the algorithm's performance. Can the third row from the bottom in Table 8 be considered as the results of TD3+BC with several techniques, including changes to $\gamma$, the use of three layers for the networks, LayerNorm, and the decoupling of actor and critic penalties? Is this why the authors state that 'ReBRAC's critic regularization does not play a significant role in the algorithm's performance' and that 'TD3+BC experiments will not provide further insights'?
> >
> > Regarding the plasticity loss, I understand how the authors computed it. Has any previous study used a similar approach? I am asking just for confirmation.

---

> ### Author Response · Authors · 2024-10-21
>
> ## About normalizations
> We treat the choice of the normalization type as a hyperparameter so that is why we put these results into appendix and reported the best one in the main text which is LayerNorm. SpectralNorm is missing from the IQL due to the fact that it performed compared to other options so we omitted it to reduce the amount of compute.
>
> ## About ReBRAC
> Yes, the core idea behind ReBRAC is that small changes that were introduced along with various approaches (e.g. deeper critics or LayerNorm) might significantly affect the performance. ReBRAC is actually TD3 + BC with those small adjustments which notably boosted it's performance. There is also a critic penalty which is brought by origin of TD3 + BC -- BRAC. However, as you already mentioned in Table 8 of ReBRAC paper authors demonstrate that this change does not play a significant role and that is why we state it in our response.
>
> ## About plasticity
> To our prior knowledge, plasticity was not tested like that before. However, we hope that our reasoning behind the approach we use is convincing enough.

---

> > ### Comment · Reviewer_ZAs5 · 2024-10-29
> >
> > - Concerning plasticity, I believe the authors' approach to quantifying it may not be appropriate. Even if the policy model has comparable plasticity, a policy closer to the behavior policy from the start would result in a lower BC loss. Additionally, the variance in the BC loss term seems too high to observe meaningful differences between the variants.
> >
> > - In DroQ (Hiraoka et al., 2021), Layer Normalization is applied after Dropout. However, in this work, the authors used Layer Normalization before Dropout. Was this difference intentional or accidental? Did the authors conduct any preliminary experiments to examine the order of Layer Normalization and Dropout?

---

> > > ### Author Response · Authors · 2024-10-30
> > >
> > > 1. We assume that our approach should address the main goal of the plasticity, i.e. ability to learn another function. We run 100 updates with the batch size of 1024 which means that each policy is updated with 100k examples. This amount is quite big and on practice we see that it reveals some effects, e.g. ReBRAC with dropout has notably less plasticity (and worse performance) than other variants of ReBRAC and error bars do not intersect.
> > > 2. Our choice is accidental. We did not check the effect of the order but we agree that it might play some role.

---

> > > > ### Comment · Reviewer_ZAs5 · 2024-10-31
> > > >
> > > > >  ReBRAC with dropout has notably less plasticity (and worse performance) than other variants of ReBRAC and error bars do not intersect.
> > > >
> > > > Which figure are the authors referring to? I think the error bars overlap in many cases.

---

> > > > > ### Author Response · Authors · 2024-10-31
> > > > >
> > > > > We refer to figures in appendix I

---

> > > > > > ### Comment · Reviewer_ZAs5 · 2024-10-31
> > > > > >
> > > > > > The BC loss is significantly higher for ReBRAC with Dropout only in the AntMaze task, while the error bars overlap in the other tasks. Is that correct?

---

> > > > > > > ### Author Response · Authors · 2024-10-31
> > > > > > >
> > > > > > > That is correct and it correlates with significant performance drop on those tasks when the dropout is used

---

### Author Response · Authors · 2024-10-19

We updated the script based on the reviewers' feedback

---

### Decision · Action_Editor_JD1e · 2024-11-21

**Recommendation:** Reject

**Comment:**

This paper is an empirical study of regularization in actor networks in offline RL. The authors experimented with two RL algorithms and showed that various combinations of standard regularization techniques lead to improvements.

The reviewers were excited about this paper and but had several concerns:

* Two reviewers asked the authors to run additional experiments. There requests were turned down, supposedly because the authors do not have access to the assets anymore. While I sympathize with this situation, this cannot be used as a reason not to scrutinize a purely empirical paper.

* Reviewer ZAs5 raised a concern that measuring policy plasticity using a behavior cloning loss is wrong. The authors did not take this comment into account. The reviewer is concerned that if this paper is accepted, subsequent papers in this field may adopt the same methodology.

* Reviewer Nmm2 does not recommend acceptance of the paper because the authors refused to take even simple suggestions into consideration. The reviewer understands that the authors cannot run more experiments. However, even suggestions to examine reward sequences from the existing result, or add one or two paragraphs to position the paper better, were not taken seriously.

Given these concerns, the paper cannot be accepted. I encourage the authors to work closer with their reviewers next time.

**Audience:**

Yes. Offline RL is a popular topic and regularization of actors is understudied.

**Claims And Evidence:**

No. This paper is an empirical study of regularization in actor networks in offline RL. While it is comprehensive, the reviewers had several concerns and suggested changes. Some suggestions were not taken seriously.

**Resubmission Of Major Revision:**

The authors may consider submitting a major revision at a later time.